# Liquid-state quantitative SERS analyzer on self-ordered metal liquid-like plasmonic arrays

Li Tian[1], Mengke Su[1], Fanfan Yu[1], Yue Xu[1,2], Xiaoyun Li[3], Lei Li[1], Honglin Liu [1,2] & Weihong Tan[2,4]

Liquid interfacial plasmonic platform is emerging for new sensors, catalysis, and tunable optical devices, but also promises an alternative for practical applications of surface-enhanced Raman spectroscopy (SERS). Here we show that vigorous mixing of chloroform with citrate-capped gold nanorod sols triggers the rapid self-assembly of three-dimensional plasmonic arrays at the chloroform/water (O/W) interface and produces a self-healing metal liquid-like brilliant golden droplet. The O phase itself generates stable SERS fingerprints and is a good homogeneous internal standard for quantitative analysis. This platform presents reversible O/W encasing in a common cuvette determined just by surface wettability of the container. Both O-in-W and W-in-O platforms exhibit excellent SERS sensitivity and reproducibility for different analytes by the use of a portable Raman device. It paves the way toward a practical and quantitative liquid-state SERS analyzer, likened to a simple UV–Vis spectrometer, that is far superior to typical solid substrate-based or nanoparticle sol-based analysis.

[1] College of Food Science and Engineering, Hefei University of Technology, Hefei, Anhui 230009, China. [2] Molecular Science and Biomedicine Laboratory, State Key Laboratory of Chemo/Bio-Sensing and Chemometrics, College of Chemistry and Chemical Engineering, College of Life Sciences, and Aptamer Engineering Center of Hunan Province, Hunan University, Changsha, Hunan 410082, China. [3] Shanghai Synchrotron Radiation Facility, Shanghai Institute of Applied Physics, Chinese Academy of Sciences, Shanghai 201204, China. [4] Department of Chemistry and Department of Physiology and Functional Genomics, Center for Research at the Bio/Nano Interface, Health Cancer Center, UF Genetics Institute, McKnight Brain Institute, University of Florida, Gainesville, FL 32611-7200, USA. These authors contributed equally: Li Tian, Mengke Su. Correspondence and requests for materials should be addressed to H.L. (email: liuhonglin@mail.ustc.edu.cn) or to W.T. (email: tan@chem.ufl.edu)

Biochemical sensing calls for the development of analyzers capable of precision identification of various targets in complex media[1]. Compared with traditional chromatography/mass spectrometry[2], the surface-enhanced Raman spectroscopy (SERS) technique has proved to be an efficient screening method with single-molecule sensitivity[3]. The rapid emergence of novel nanomaterials and various Raman devices has accelerated the wide application of SERS in analytical sciences[4]. Particularly, anisotropic nanoparticles, like gold nanorods (GNRs), possess superiority in structural self-assembly and light-harvesting capacity[5]. The surface plasmon resonance (SPR) features of GNRs can be easily tuned from visible to infrared spectrum to match with the laser wavelength of portable Raman devices, e.g. commonly used 785 nm[6]. However, quantitative SERS analysis faces significant challenges because this near-field phenomenon is mainly attributed to intense electromagnetic enhancement within nanoscale plasmonic gaps and greatly affected by detailed local nanostructures[7].

Typical SERS analysis mainly involves solid substrate-based analysis and nanoparticle sol-based analysis[8,9], but faces difficulties in controlling the uniformity of nanoscale hot spots and the inefficiency of placing the targeted molecules in prefabricated hot spots. Few substrates can simultaneously achieve both high sensitivity and high reproducibility. Naturally, internal standard (IS) tags have been employed to calibrate SERS fluctuation[10]. However, discrete IS tags, just like targeted analytes, face the similar localization problem on nanoscale metal surfaces[11]. Even if IS and analytes could be uniformly distributed on the metal surface, either the competing adsorption or dynamic exchange on the metal surface would still exist, especially in multiphase systems and complex samples.

Mirror-like properties of two-dimensional (2D) arrays of gold nanospheres at the liquid–liquid interface[12] imply an impressive capacity of this new kind of plasmonic platform for tunable optical devices[13], sensors[14], and catalysis[15,16]. These platforms have also been used to achieve multiplex SERS detection in multiple phases with high sensitivity[17–19]. Interfacial liquid-state 2D array is variable, versatile, and self-healing, provides a molecularly sharp and defect-free focal plane[20,21], and virtually guarantees the stable controlling of nanogap sizes and the feasible localizing of

analytes[18,22]. However, SERS measuring on horizontal liquid-state arrays involves a very fundamental tradeoff between precise laser focusing and light-path arrangement[19,23], which encounters fluctuations in liquid levels, especially for portable Raman devices.

Liquid interfacial self-assembly usually requires the assistance of salts, inducers, promoters, or accelerators for structural aggregating[24,25] or altering the surface charges to reduce the Coulomb repulsion[26–31]. Three-dimensional (3D) silver colloidal superstructures were assembled through mixing cyclohexane/water under emulsified conditions[32], but liquid-state tiny emulsions have poor SERS performance and their dried films face similar dilemma to solid substrate-based analysis. Recently, the reflective gold nanosphere arrays encasing a macroscopic droplet, denoted as metal liquid-like droplets[28], were prepared with the assistance of lipophilic promoters. But the surfactants generally hinder the entrance of analytes into active sites and consequently interfere with Raman enhancement. Developing a substrate-free and clean 3D liquid-state platform for quantitative SERS analysis promises a higher capability than 2D horizontal arrays and allows more reliable light-path arrangement[29].

Here we show a reversible O/W encasing strategy to self-assemble metal liquid-like 3D GNR arrays just by controlling the surface wettability of the containers (Fig. 1). CTAB-capped GNR (Ct-GNR) sols are modified into citrate-capped GNR (Ci-GNR) sols which enable a cleaner plasmonic surface and long-term stability[30,33]. The Ci-GNRs are able to quickly self-assemble at the O/W interface and make a 3D large-scale metal liquid-like droplet. The self-assembly is triggered by adding chloroform into the Ci-GNR sols without other inducers or modifiers. This 3D plamonic platform is self-healing, easy to operate, and needs no engineering. Surprisingly, the O phase itself generates stable SERS fingerprints that can be used as a good homogeneous IS tag for quantitative SERS analysis on a portable Raman device, and the ratiometric SERS intensity of target molecules is generated in a stable, continuous, and predictable manner. Moreover, the O phase acts as an extraction solvent to separate and concentrate the oil-soluble targets from complex media. This platform exhibits excellent multiplex and multiphase sensing capability evidenced by dual-analyte detection.

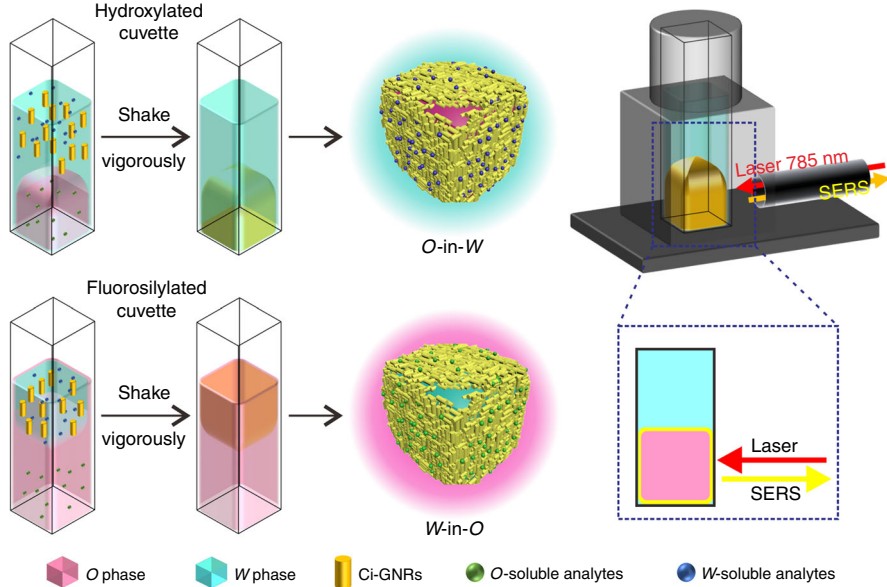

**Fig. 1** A multiphase liquid-state SERS analyzer. Reversible O/W encasing for self-assembly of metal liquid-like GNR arrays is realized in a common cuvette. Detailed experimental setups are shown in Supplementary Fig. 1

## Results

**GNR surface modification**. The Ci-GNR sols were prepared through a ligand exchange method (Fig. 2a). First, homogeneous GNRs with a length–diameter ratio of 3.0 were successfully synthesized using CTAB as the capping agent (Fig. 2b)[31]. The SPR of Ct-GNR sols (Fig. 2e) generated two main bands at 521 and 705 nm, corresponding to the transverse SPR and the longitudinal SRP, respectively. However, CTAB adsorption on the GNR surfaces forms micelles in aqueous solution and prevents aggregation via DLVO mechanisms[34]. This feature limits the self-assembly of GNR arrays at the liquid/liquid interface.[35,36] The poly(4-styrenesulfonic acid) (PSS) molecule, as the detergent and mediation agent[30,37], was used to replace CTAB surfactants on the GNR surface because of its preferential accumulation of higher molecular weight substances on the solid surface. The PSS molecule adsorbs weakly onto GNR surfaces in the absence of CTAB; nevertheless, PSS-capped GNR (Pss-GNR) sols could be obtained within a short time. The LSPR band in the UV–Vis absorbance spectrum of Pss-GNR sols presented a slight blueshift of ca. 2 nm (Fig. 2f), indicating morphological alteration after surfactant exchange. However, the relatively narrow width of the LSPRs implied that no aggregation had occurred. Transmission electron microscope (TEM) observation evidenced that most GNRs changed into a dog bone shape after having replaced the CTAB (Fig. 2c), which was consistent with the UV–Vis spectra.

Next, citrate as the final stabilizing agent was used to replace PSS through a similar surfactant exchange in a process of repeated centrifugation and redispersion. The Ci-GNR sols displayed features very similar to those of Pss-GNR sols in the UV–Vis spectrum (Fig. 2g, black line). TEM observations evidenced a much more uniform dog bone-shaped GNR (Fig. 2d). Importantly, the aged Ci-GNR sols could be stored at 4 °C for quite a long time, and a nearly identical UV–Vis spectrum was obtained after storage for 1 week (Fig. 2g, red line). In other words, dispersion stability is retained after surfactant exchange from CTAB to citrate. In the following studies, this kind of Ci-GNR with relatively clean surface demonstrated broad capability in self-assembly of GNR arrays, especially for liquid/liquid interfacial self-assembly. Moreover, Ci-GNRs without any other surfactants provide a simple analysis environment.

Conventional synthesis of GNR sols commonly produces a significant fraction of thermodynamically favorable irregular structures, such as dog bone-like structures[35]. The main reason is the preferential binding of CTAB molecules to the middle of the nanorods, resulting in the deposition of more gold atoms at the ends[5]. Ligand exchange is predicated on employing a molecule able to displace CTAB[38] and ensure its tight binding to the gold surface under different solution conditions[39]. During our process of displacing CTAB by other ligands, the prolonged centrifugation and redispersion might alter the surface structure of GNRs to some extent. Also, the uneven capping of CTAB and subsequent polymer or citrate could induce atomic rearrangements of GNRs and produce the aforementioned dog bone-like shape.

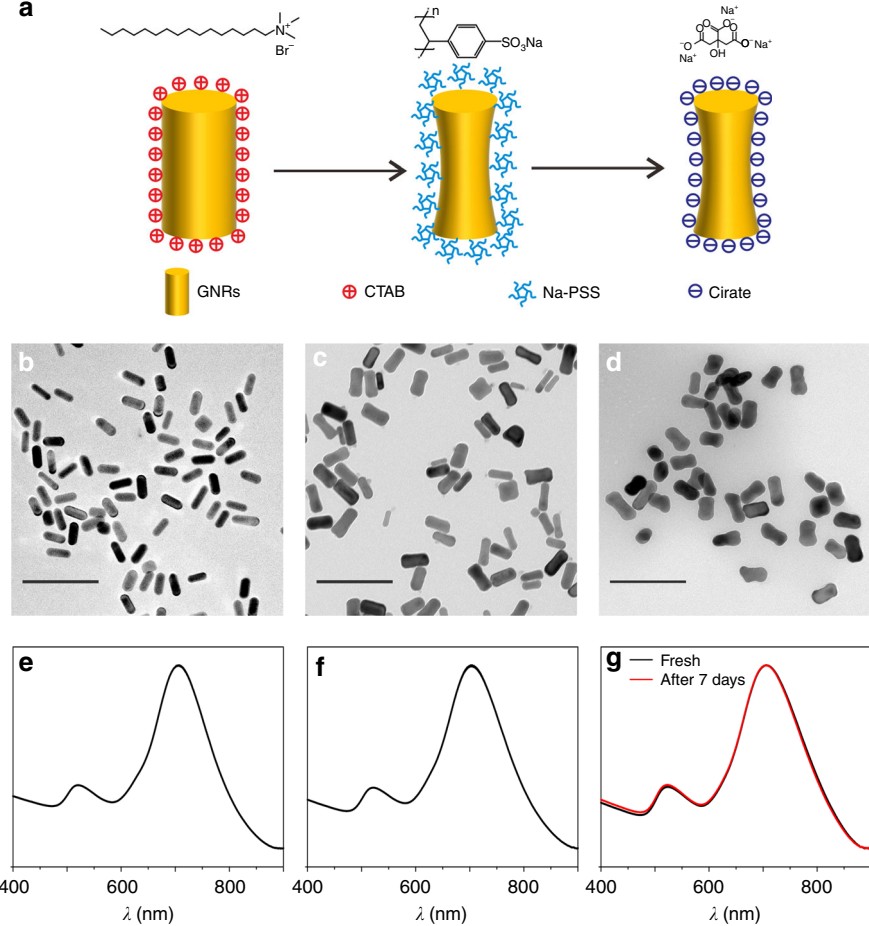

**Fig. 2** GNR surface modification. **a** Schematic illustration of the PSS-mediated surface modification from CTAB to citrate. TEM observations and the corresponding UV–Vis absorbance spectra of the as-prepared Ct-GNR sols (**b**, **e**), Pss-GNRs sols (**c**, **f**), and Ci-GNR sols (**d**, **g**). A UV–Vis spectrum of the Ci-GNR sols aged for 1 week (red line) was overlapped in **g** for contrast. All of the scale bars are 200 nm

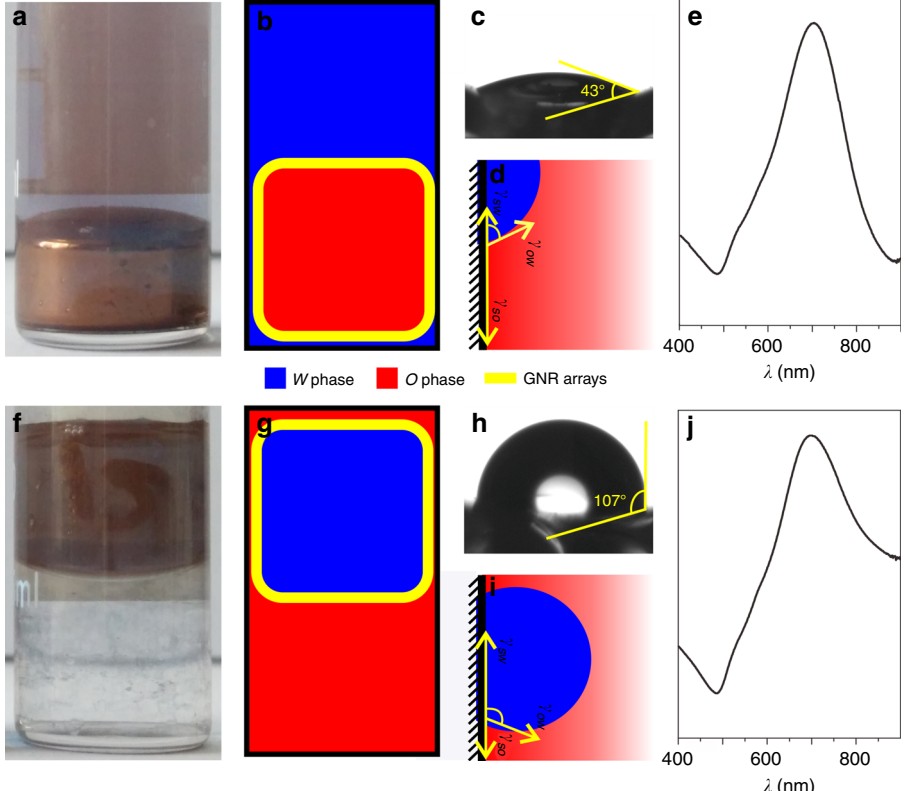

**Fig. 3** Metal liquid-like self-ordered GNR arrays. **a** Optical image and **b** schematic of *O*-in-*W* encasing, **c** the corresponding water contact angle test, and **d** the hypothesized coexistence of *S/O/W* triple-phase boundary in a hydroxylated cuvette. **f** Optical image and **g** schematic of *W*-in-*O* encasing, **h** the corresponding water contact angle test, and **i** the hypothesized coexistence of *S/O/W* triple-phase boundary in fluorosilylated cuvette. **e**, **j** UV–Vis absorbance spectra of GNR arrays on *O*-in-*W* and *W*-in-*O* interfaces, respectively

**Reversible *O/W* encasing and surface wettability effect**. Just by adding 1 mL of chloroform into equivoluminal Ci-GNR sols, a brilliant golden film covering the *O* phase was successfully fabricated on the *O/W* interface, emerging as metal liquid-like 3D GNR arrays without using any promoter. The volume of both *O* and *W* phases could be adjusted after the formation of interfacial arrays. This step could increase particle density at the *O/W* interface. The successful dense GNR encasing of *O* phase mainly depends on the hydrophilic internal surface of the glass cuvette (Fig. 3a, b) and its hydroxylation treatment by fresh piranha solution generated a contact angle of 43° (Fig. 3c). This accidental finding led us to ask what would happen in the event of hydrophobic treatment of glass cuvette surface. Surprisingly, a completely inverse encasing of GNRs on the *W* phase was achieved merely through the hydrophobic fluorination treatment of the cuvette's surface (Fig. 3f, g), demonstrating a water contact angle of ca. 107° (Fig. 3h). The UV–Vis absorption maximum of interfacial Ci-GNR arrays redshifted to 765 nm in the hydrophilic container (Fig. 3e) and 723 nm in the hydrophobic container (Fig. 3j), respectively. The redshift could be attributed to dense self-assembling GNR arrays and strong interparticle plasmonic coupling effect, and its difference in two situations could be attributed to the change of the dielectric environment around the GNR arrays in different encasings. An even more interesting phenomenon occurred with the reversible *O/W* encasing when pouring the same assembly system out of the hydrophilic cuvette and into the hydrophobic cuvette. We found that this resulted in the induction of GNR encasing of the *W* phase and, conversely, the induction of GNR encasing of the *O* phase. Supplementary Movie 1 presented three cycles of reversible *O/W* transition of metal liquid-like GNR arrays.

We once tried to produce these metal liquid-like interfacial arrays by the direct use of CTAB-capped GNRs. Emulsification occurred after CTAB-capped GNR sol was oscillated by vortex for a period of time. However, the emulsions could not fuse into a single globule, possibly because the bilayer micelle of CTAB molecules increased steric hindrance, thereby impeding GNR accumulation onto the *O/W* interface. In fact, the self-assembly of CTAB-capped GNRs only involves 2D horizontal arrays on planar liquid/liquid or solid/air interface[19]. Nonetheless, this behavior appears to be concentration-dependent, making the assembly morphology and interparticle gap hard to control as a consequence of the uncertain number of CTAB micelle layers[39,40]. Taking these facts into consideration, we speculated that the assembly of Ci-GNRs might rely on spontaneous diffusion-limited nanoparticle localization to the *O/W* interface. The emulsification process has a key role in reducing the average distance between the nanoparticles and the *O/W* interface, thereby speeding up diffusion-limited localization to the interface[18].

These interesting phenomena inspired us to perform a theoretical analysis. Chloroform, i.e. the *O* phase, is slightly heavier than water, i.e. the *W* phase. In the mixed system, the buoyancy approximately equals the gravity of the *O* phase. Irrespective of other factors, e.g., surface roughness, we assumed an initially balanced contact angle at the solid/oil/water (*S/O/W*) triple-phase boundary (Fig. 3d), considering a thermodynamic equilibrium among the three phases: solid (*S*), organic (*O*), and immiscible water (*W*). The shape of this interface is determined by the Young–Laplace equation:[36]

$$\gamma_{so} = \gamma_{sw} + \gamma_{ow}\cos\theta, \qquad (1)$$

where $\theta$ is the contact angle, $\gamma_{so}$ is the $S/O$ interfacial energy (i.e., the surface tension), $\gamma_{sw}$ is the $S/W$ interfacial energy, and $\gamma_{ow}$ is the $O/W$ interfacial energy. Zhu et al.[41,42] have derived the numerical method for liquid–solid interfacial wettability as

$$\gamma_{sw} = \frac{\gamma_{ow}}{2}(\sqrt{1 + \sin^2\theta} - \cos\theta),$$

$$\gamma_{so} = \frac{\gamma_{ow}}{2}(\sqrt{1 + \sin^2\theta} + \cos\theta). \qquad (2)$$

Therefore, when $\theta$ is less than 90°, $\gamma_{so}$ is greater than $\gamma_{sw}$. In other words, the $W$ phase will advance, and the $O$ phase will recede along the $S$ surface, finally inducing the disappearance of the triple-phase boundary such that the $W$ phase will quickly spread along the cuvette surface and encase the entire $O$ phase. In contrast, $\gamma_{sw}$ is greater than $\gamma_{so}$ when $\theta$ is larger than 90°, and the $W$ phase will recede, while the $O$ phase advances along the surface, also inducing the disappearance of the triple-phase boundary. Similarly, the $O$ phase will inversely encase the entire $W$ phase. In both situations, the surface tension, rather than the buoyancy or gravity of liquids, dominates the whole liquid distribution on the solid surface. SERS measurements by focusing the laser on the top interfacial GNR film of $W$-in-$O$ platform produced clear Raman fingerprint peaks of chloroform (Supplementary Fig. 2), demonstrating the existence of a chloroform layer on the top of water phase (Fig. 3g). This is indirect proof of surface tension-driven liquid distribution. Additionally, vigorous shaking, or, alternatively, ultrasonication, effectively facilitated the spontaneous adsorption of individual nanoparticles at the interface[28]. The driving force behind this spontaneous interfacial adsorption is the diminution of excess surface energy at an early stage of the Ci-GNRs. Vortex mixing creates shear forces that can be particularly strong at $O/W$ interfaces, making it easier for Ci-GNRs to reach the liquid boundary and localize onto the $O/W$ interface because of $O/W$ interfacial tension. Furthermore, it is likely that the interparticle van der Waals forces make a significant contribution to the formation of the GNR film on the $O/W$ interface.

**Self-assembly structure of GNR array at $O/W$ interface**. The exact structure of GNR layers at the $O/W$ interface is extremely difficult to assess[18]. Here, a conventional dark-field microscopy (DFM) image of diluted GNRs on the $O/W$ interface displays discrete particles with various colors (Fig. 4a, b), but red is dominant, indicating the dominance of longitudinal SPR band of GNR on the $O/W$ interface, i.e., most GNRs have horizontal orientation on the $O/W$ interface. Similar results were obtained with different particle densities of GNR on the $O/W$ interface (Supplementary Fig. 3). Although DFM images do not exclude the influence of nanorod aggregates, the speculation of horizontal orientation of GNR on $O/W$ interface is consistent with SEM observations. Evidence of horizontal orientation of GNRs was further demonstrated by SEM observations of dried assembly array on a silicon wafer. Indeed, under the condition of close-packed GNR arrays on $O/W$ interface, the dried film on silicon wafer did present horizontal orientation of all GNRs (Fig. 4c). Considering the diffusion effect, variable particle densities of GNR on the $O/W$ interface were examined, but all of the dried films presented horizontal orientation of GNRs in either sub-monolayer or multilayer films (Supplementary Fig. 3). Hence, we consider a horizontal orientation of GNRs on $O/W$ interface. If not, the dried array should have vertical GNRs or at least fall down in preferential direction under close-packed conditions.

The number of GNR layers on $O/W$ interface was further observed. At low concentration of GNRs, the array arrangement is single layer and has big gaps (red arrowheads in Supplementary Fig. 3). As the number of GNRs increases, the interparticle spaces gradually decrease. At higher concentration, the GNR arrangement appears multilayered, likely a result of solvent evaporation, and the interparticle spaces could not be further decreased because of the energy barriers at $O/W$ interface where excess GNPs tend to be arranged in multilayers. This inference is consistent with optical observations that the single-layer or subsingle-layer assembly of GNPs on $O/W$ interface produces a metal liquid-like brilliant golden droplet, but that much higher concentrations cause surface wrinkling and loss of metallic luster.

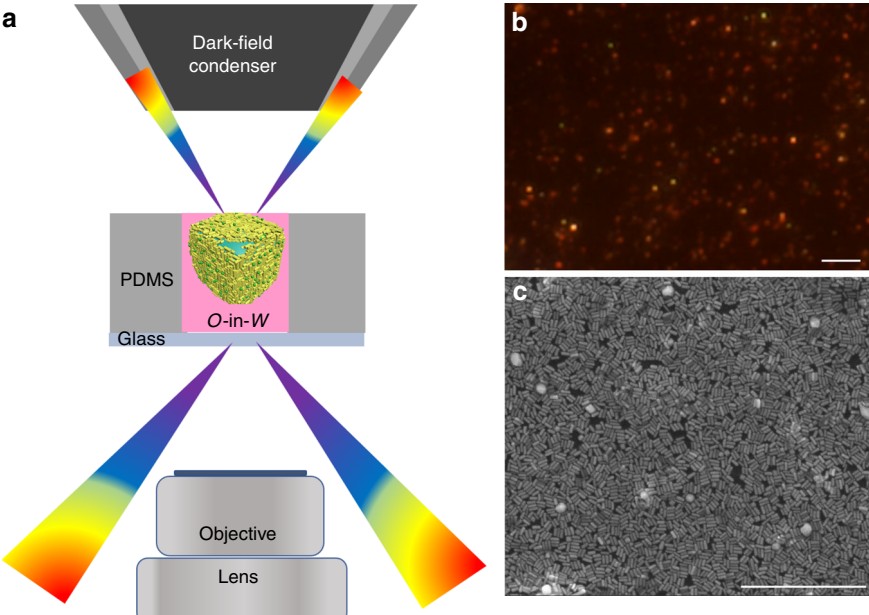

**Fig. 4** Interfacial morphology of GNR arrays. **a** Dark-field imaging setup for metal liquid-like $O$-in-$W$ GNR arrays in PDMS cavity. **b** DFM image of interfacial GNRs at 0.6 OD of 1 mL sols. The scale bar is 4 μm. **c** SEM image of close-packed GNR film dried on silicon wafer fabricated from 1 mL of 7 OD GNR sols. The scale bar is 1 μm

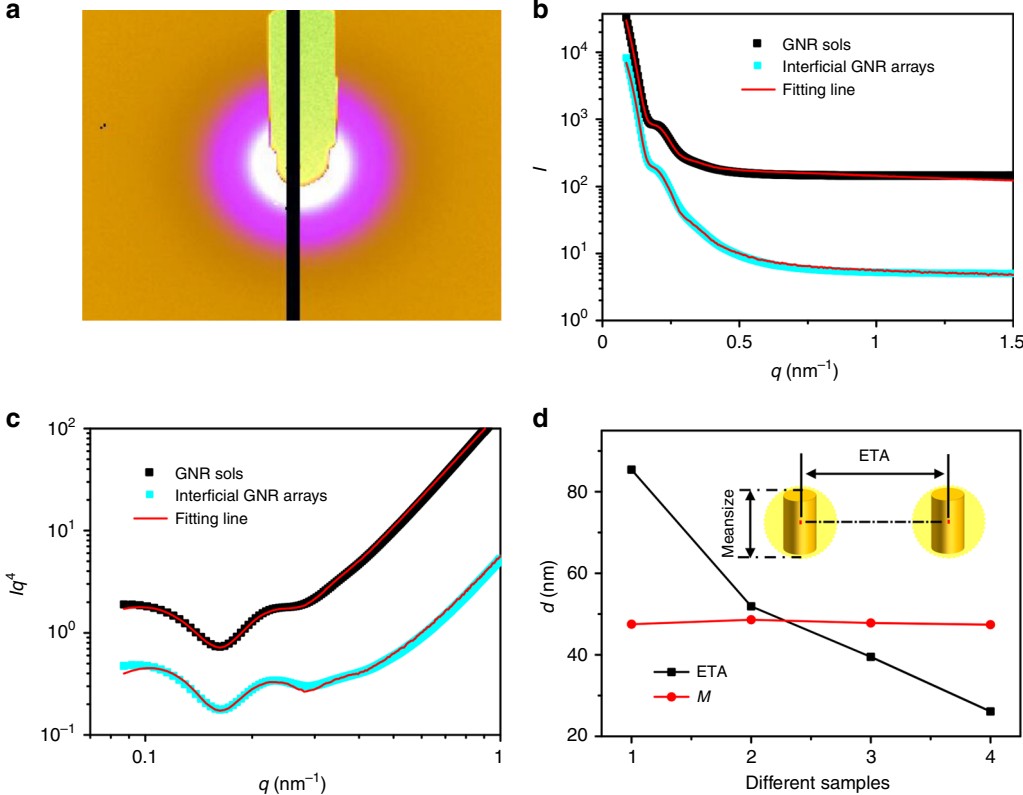

**Fig. 5** SR-SAXS examinations on interfacial structures of GNR arrays. **a** A typical 2D SR-SAXS pattern. **b** 1D SR-SAXS scattering data and fitting lines of GNR sols and interfacial GNR arrays, respectively, plotted as SR-SAXS intensity ($I$) vs. scattering vector modulus ($q$). **c** Transformed SR-SAXS curves in **b**, plotted as $q^4 \times I$ vs. $q$. **d** Calculated average scattering size ($M$) and center-to-center distance (ETA) of GNRs from different samples: 1, as-synthesized GNR sols of 3.0 OD; 2–4, interfacial arrays fabricated by GNR sols with varied OD values: 3.0, 6.0, and 9.0, respectively

More importantly, in situ SR-SAXS measurements were performed to confirm the information on orientation and relative distance of GNRs array at liquid–liquid interface. A typical 2D SR-SAXS pattern (Fig. 5a) was integrated with 1D SR-SAXS scattering curves (Fig. 5b), and scattering intensity $I$ of the SAXS curves was transformed into $q^4I$ plotted as a function of $q$ to highlight the scattering peaks of GNRs (Fig. 5c). According to the theoretical fitting (Supplementary Methods), the calculated average scattering size ($M$) of GNRs at O/W interface is 47.8 nm (Fig. 5d), consistent with TEM observations, and that the distribution of particle sizes (Supplementary Fig. 4) implies the discrete states of GNRs rather than the aggregates after self-assembly on O/W interface. In addition, the ETA values gradually decreased from 85.4 to 26.1 nm, i.e., the interparticle space decreased down to negative values at larger GNR densities. Under limited conditions, it should be noted that ETA is a fitting parameter for which the tendency toward relative changes is more important than numerical magnitudes. As mentioned above, high concentrations of GNRs at O/W interface did not induce interparticle aggregation; hence, the decreased ETA values at high GNR concentrations could be attributed to the multilayer arrangements of GNRs at O/W interface. The negative values of interparticle space can be explained by the effect of GNRs in multilayer arrangements which would have no interparticle space for X-ray penetration in physical. The results, when combined with SEM and DFM observations, indicated that close-packed arrays of GNR multilayers were formed under high concentrations of GNRs. Another key issue is the relationship between layer number and SERS enhancement at O/W interface that will be discussed later.

**Reliable IS tags for quantitative SERS analysis**. Chloroform itself on the interfacial GNR arrays generates stable SERS signals with a sharp fingerprint peak at 662 cm$^{-1}$ (Supplementary Fig. 5). The stability of this characteristic peak can be attributed to C–Cl symmetric vibration. Therefore, it could be used as an inherent IS tag to calibrate the spectral intensity of targeted analytes. To examine the feasibility of this idea, we chose TBZ, a broad-spectrum oil-soluble fungicide without resonance Raman effects, as the targeted analyte.

Chloroform containing TBZ of different concentrations was used to fabricate the GNR arrays on an O-in-W interface in a hydrophilic cuvette (Fig. 6). The generated SERS spectra presented clear fingerprint peaks of TBZ (Supplementary Fig. 6). The peaks at 1006, 1272, 1540, and 1571 cm$^{-1}$ could be assigned to the out-of-plane bending of C–C–C ring stretching, C=N stretching, C–C stretching, and ring deformation vibration, respectively, while the peak at 780 cm$^{-1}$ was assigned to the out-of-plane C–H bending (Supplementary Table 1)[43]. The peak of 0.05 ppm TBZ could be clearly distinguished, and the peak strength rose steadily with increasing molecular concentrations (Fig. 6a).

Meanwhile, a significant peak at 662 cm$^{-1}$ originating from chloroform was observed, and its strength was very stable under any TBZ concentration. The two peaks at 780 and 662 cm$^{-1}$ assigned to TBZ and chloroform, respectively, are very close, but distinguishable. Consequently, we used them to generate a ratiometric indicator ($r$) of relative peak intensity, $r_{780/662}$, for quantitative analysis of TBZ. Indeed, a good linear relationship was plotted between $r_{780/662}$ and logarithmic concentration of TBZ over a large range of 0.05–1000 ppm with a correlation

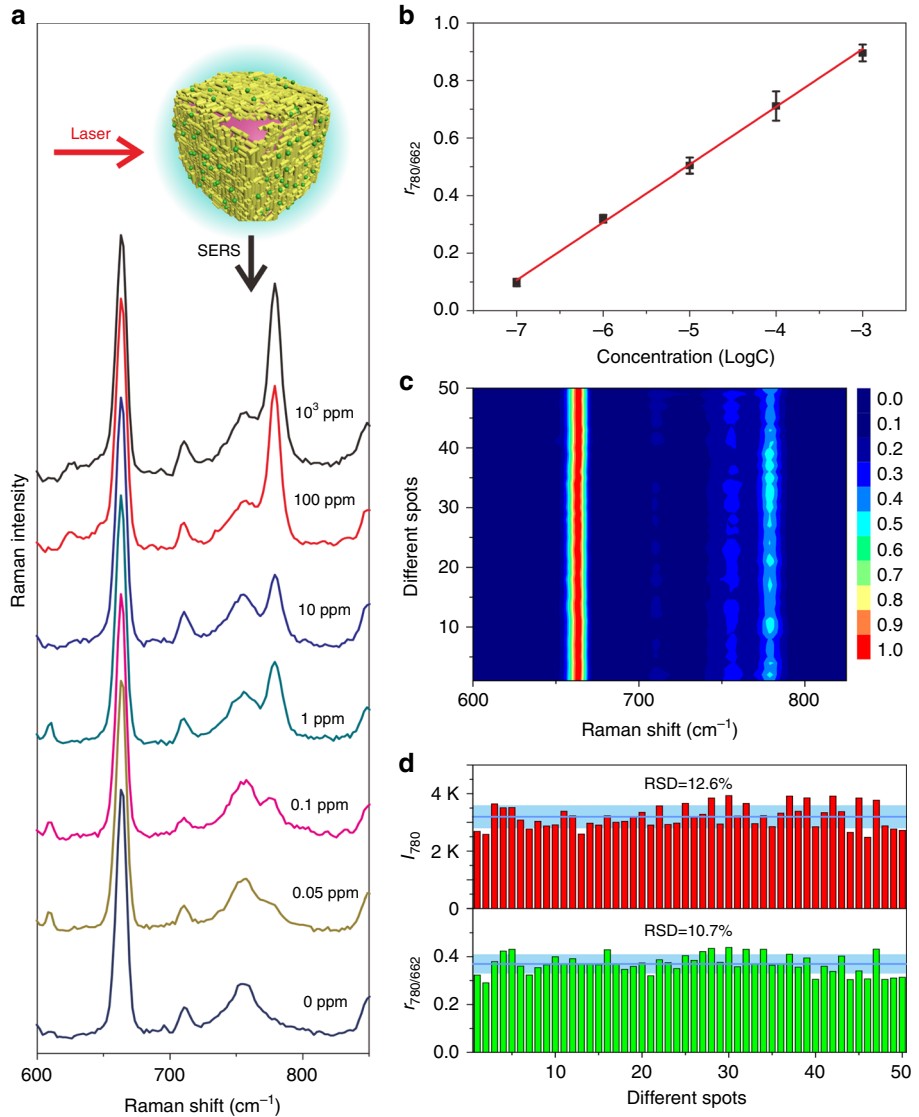

**Fig. 6** Metal liquid-like GNR arrays on *O*-in-*W* interface for SERS analysis of TBZ. **a** SERS spectra of TBZ with concentrations of 0, 0.05, 0.1, 1, 10, 100, and 1000 ppm, respectively, dissolved in the *O* phase. **b** A linear relationship between the $r_{780/662}$ values and logarithmic TBZ concentration. **c** Fifty random runs in triplicate SERS experiments generating a 2D spectral mapping of 10 ppm TBZ. **d** Statistical histograms of $r_{780/662}$ and $I_{780}$ values, respectively. The error bars represent the statistical RSD and were calculated from five different runs

coefficient ($R^2$) of 0.997 (Fig. 6b). Fifty runs in triplicate SERS experiments on GNR arrays generated a visual 2D spectral mapping by using 10 ppm TBZ (Fig. 6c), and the relative standard deviations (RSD) of $r_{780/662}$ and $I_{780}$ were 10.7% and 12.6%, respectively, indicating good improvement of signal reproducibility (Fig. 6d).

Since the *O/W* encasing is reversible, we further examined the performance of IS tags in *W*-in-*O* interface in a hydrophobic cuvette (Supplementary Fig. 7). TBZ of different concentrations was also dissolved in the *O* phase. Interestingly, the interfacial GNR arrays on *W*-in-*O* interface produced similar spectral behaviors, and 0.1 ppm of TBZ well presents its fingerprint peak at 780 cm$^{-1}$. Again, a good linear relationship was plotted between $r_{780/662}$ and logarithmic concentration of TBZ over the range of 0.1–1000 ppm with a $R^2$ of 0.978. Fifty runs in triplicate SERS experiments generated a visual 2D spectral mapping by using 10 ppm TBZ, and the RSD of $r_{780/662}$ and $I_{780}$ was 17.1% and 26.9%, respectively. Chloroform molecules in this platform are homogeneously filled in the nanogaps of GNP arrays and can tolerate the coexistence of

targeted molecules. The results demonstrate that the *O* phase as IS tags could greatly improve SERS signal reproducibility.

The fingerprint peaks of TBZ were the same in both *O*-in-*W* and *W*-in-*O* platforms (Supplementary Table 1), indicating a similar sensing behavior of metal liquid-like GNR arrays. However, the *O*-in-*W* platform produced a better limit of detection (LOD) and slightly smaller RSD values than the *W*-in-*O* platform. This difference might originate from the higher volatility of chloroform compared with water. In the case of *W*-in-*O* encasing, the rapid volatilization of external *O* phase creates a noticeable disturbance, not only to the GNR arrays, but also to the targeted molecules. UV–Vis spectrum of GNR arrays on the *O*-in-*W* interface has a sharper SPR coupling band (Fig. 3e), while that on the *W*-in-*O* interface shows larger absorbance at longer wavelengths (Fig. 3j), indicating reduced uniformity. These results are consistent with the inference of chloroform evaporation disturbance. In contrast, a much lower evaporation of water under *O*-in-*W* encasing provided a much more stable interfacial microenvironment, contributing to better signal stability and sensitivity.

We further examined the target dissolved in the $W$ phase for liquid-state SERS analysis (Supplementary Fig. 8). A water-soluble illegal ingredient, MG, typically used for the preservation of fish, was used for quantitative SERS analysis at ultra-trace levels. The interfacial platform generated clear fingerprint peaks of 0.05 ppb MG (Supplementary Fig. 9) and the observed primary peaks were the same to the assignments in literatures[44]. The normalized $r_{1612/662}$ was plotted as a function of MG logarithmic concentration, and a good linear relationship was obtained in the concentration range of 0.05–10 ppb with an $R^2$ value of 0.960 and LOD down to 0.1 ppb. Fifty repeated runs for 0.5 ppb MG generated a good 2D spectral mapping, and the RSD values of $r_{1612/662}$ and $I_{1612}$ were 12.7 and 18.8%, respectively. These results indicate that the interfacial platform is uniquely capable of quantitative SERS analysis, even under ultra-trace levels of analytes dissolved in both $O$ and $W$ phases, i.e. the $O$ phase can act as a good IS tag to calibrate the fluctuations originating from both sample and microenvironment.

**Optimized SERS enhancement with variable GNR concentrations.** SERS performance closely related to nanogaps between GNRs were optimized by controlling the amount of GNR sols added in the $O/W$ system (Fig. 7a). We found that UV absorption strength of metal liquid-like GNR arrays increased with the increase of GNR concentrations and that the absorbance maximum of interfacial GNR arrays gradually redshifted from 680 to 800 nm (Fig. 7b) when the OD values of 1 mL GNR sols increased from 3.0 to 9.0, implying a gradually enhanced interparticle SPR coupling. When the concentration of 1 mL GNR sols was 7.5 OD, almost no GNRs were left in the aqueous solution after interfacial self-assembly (Supplementary Fig. 10a). A DFM image was obtained at diluted GNR sols, but it still clearly showed that nearly all GNRs had been concentrated on the $O/W$ interface (Supplementary Fig. 10b). Based on SEM images and SR-SAXS data, it was concluded that a close-packed monolayer of GNR

array will form at the GNR concentration of approximately 7.0 OD.

TBZ of 10 ppm was then used to examine SERS performance. Fig. 7c showed that SERS signals first increased quickly with increasing GNR concentrations and then reached a plateau with GNR sols of 7.5 OD, even slightly decreasing at higher GNR concentrations. However, when the concentration of GNP sols was approximately 7.0 OD, the enhancement effect of SERS was the best. Fifty runs in triplicate SERS experiments generated an RSD of $r_{780/662}$ and $I_{780}$ with values of 6.2% and 12.8%, respectively (Fig. 7d). These results demonstrate that IS tags could greatly improve signal reproducibility. The $r_{780/662}$ at 7.5 OD GNR sols was 1.63, which is 4.3 times greater than that at 3.0 OD GNR sols. Thus, by optimizing the amount of GNR added, SERS sensitivity was greatly improved. Nevertheless, with any further increase in GNR concentration, the residual GNR in the $W$ phase might adsorb part of the analytes and induce a decrease in SERS intensity. Hence, 7.5 OD of 1 mL GNR sols was used for self-assembling the optimal interfacial arrays in the following text unless otherwise specified.

**Multiphase and multiplex liquid-state SERS analysis.** Dual-analyte detection experiments were carried out on the $O$-in-$W$ interfacial platform. In multiphase analysis experiments, MG was dissolved in the $W$ phase, the R6G was dissolved in the $O$ phase, and total molar number of R6G and MG was 0.1 nmol. The dual-analyte spectra were collected with variable molar ratios of R6G in a total 0.1 nmol molecules, denoted as 1.0, 0.8, 0.6, 0.4, 0.2, and 0 (Fig. 8a). The peaks at 608, 770, 1189, 1354, and 1642 cm$^{-1}$ were assigned to R6G (curve 5:0 in Fig. 8a). In particular, the characteristic peak at 1354 cm$^{-1}$ was assigned to the symmetric modes of in-plane C–C stretching vibrations, which can be easily distinguished from the MG band at 1395 cm$^{-1}$ assigned to $N$-phenyl stretching (Fig. 8b)[44]. All of the spectra were normalized by IS tags. More specifically, the peak strength at 662 cm$^{-1}$ and

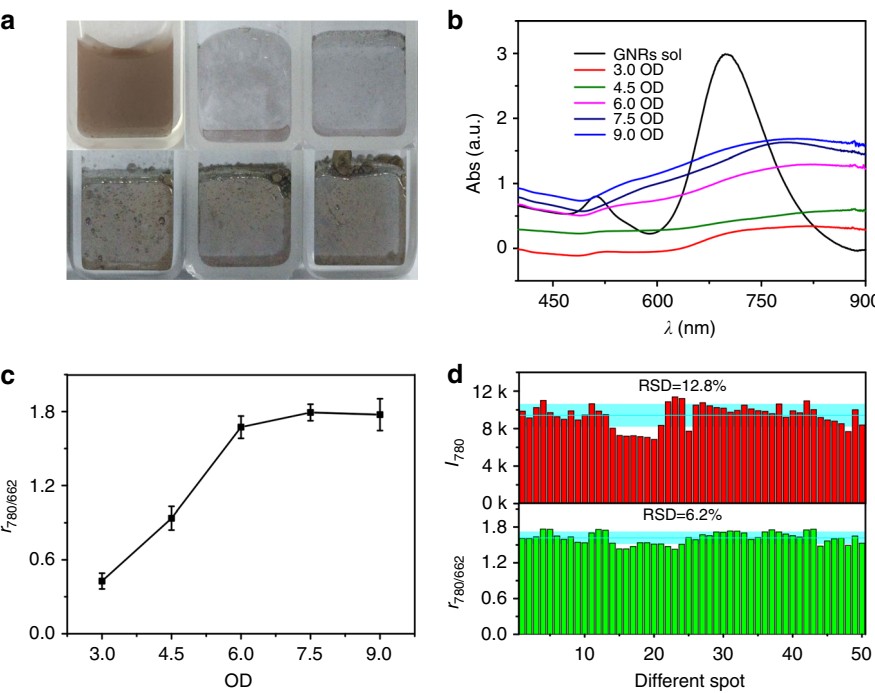

**Fig. 7** Effect of GNR concentrations on SERS performance. **a** Optical images of as-synthesized GNR sols and metal liquid-like GNR arrays on $O$-in-$W$ interface fabricated by 1 mL of GNR sols with variable OD values: 3.0, 4.5, 6.0, 7.5, and 9.0, respectively. **b** The corresponding UV–Vis absorbance spectra, **c** relative SERS strength, $r_{780/662}$, collected on interfacial GNR arrays, and **d** statistical histograms of $r_{780/662}$ and $I_{780}$, respectively. The error bars represent the statistical RSD and were calculated from triplicate SERS experiments

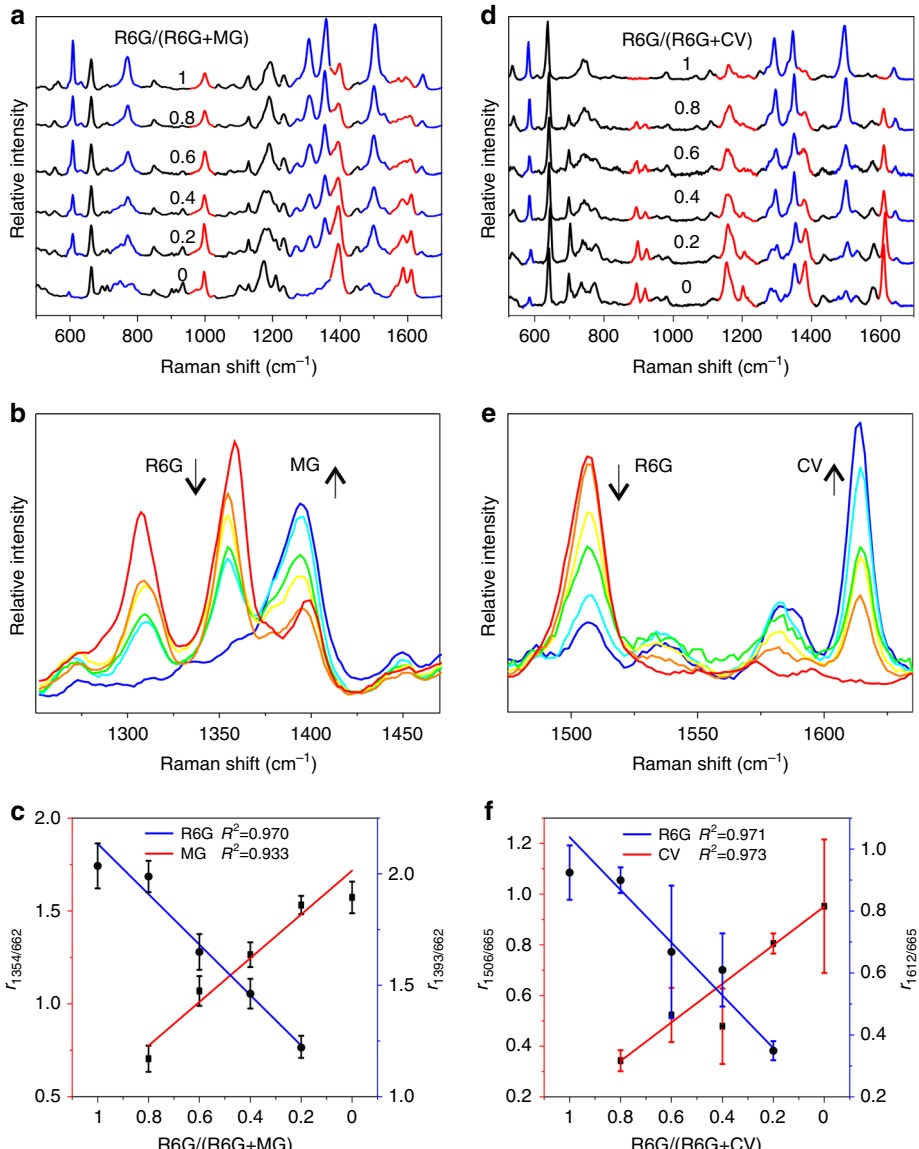

**Fig. 8** A quantitative liquid-state SERS platform. **a** Multiphase analysis of MG dissolved in the *W* phase and R6G dissolved in the *O* phase. **b** Detailed spectral variations of the 1354 cm$^{-1}$ band assigned to R6G and the 1393 cm$^{-1}$ band assigned to MG in multiphase analysis. **c** The linear relationships between the R6G/MG molar ratios and the *r* values of MG (red) and R6G (blue), respectively. **d** Multiplex analysis of R6G and CV simultaneously dissolved in the *O* phase. **e** Detailed spectral variations of the 1506 cm$^{-1}$ band assigned to RG and the 1612 cm$^{-1}$ band assigned to CV in multiplex analysis. **f** The linear relationships between the R6G/CV molar ratios and the *r* values of CV (red) and R6G (blue), respectively. Total number of molecules in each experiment was 0.1 nmol, and the molar ratio was labeled on each spectrum. The error bars represent the statistical RSD and were calculated from triplicate SERS experiments

the relative peak strengths for both R6G and MG varied regularly along with the change in molar ratios. The linear fittings were plotted on the basis of data collected in triplicate experiments and generated $R^2$ values of 0.933 and 0.970 for MG and R6G, respectively (Fig. 8c).

Similarly, in multiplex analysis experiments, a total of 0.1 nmol of CV and R6G molecules was simultaneously dissolved in the *O* phase, and the dual-analyte spectra were similarly collected with various molar ratios of R6G in a total of 0.1 nmol molecules, denoted as 1.0, 0.8, 0.6, 0.4, 0.2, and 0. All of the collected spectra were normalized by IS tags (Fig. 8d). The relative peak strengths for both R6G and CV also varied regularly along with the change in molar ratios (Fig. 8e). The fittings also indicated a good linear relationship with $R^2$ values of 0.973 and 0.971 for CV and R6G, respectively (Fig. 8f). Compared to solid substrate-based analysis,

this liquid-state SERS platform demonstrated its capability for multiphase and multiplex detection. These results demonstrated that the metal liquid-like GNR arrays for SERS analysis are reliable, reproducible, and sensitive, even under ultra-trace levels of targeted molecules.

**Toward a practical SERS analyzer**. A practical on-the-spot analyzer depends on rapid sampling and signal collection. An extraction solvent is often used to simplify sample pretreatment. TBZ analysis in pure solvent has been demonstrated in the preceding experiments. To further examine the reliability of our liquid-state SERS platform for analyzing real samples, different amounts of TBZ were spiked into fresh apple juice (Fig. 9). Since fresh juice is a complex media, SERS signals of TBZ samples without pretreatment

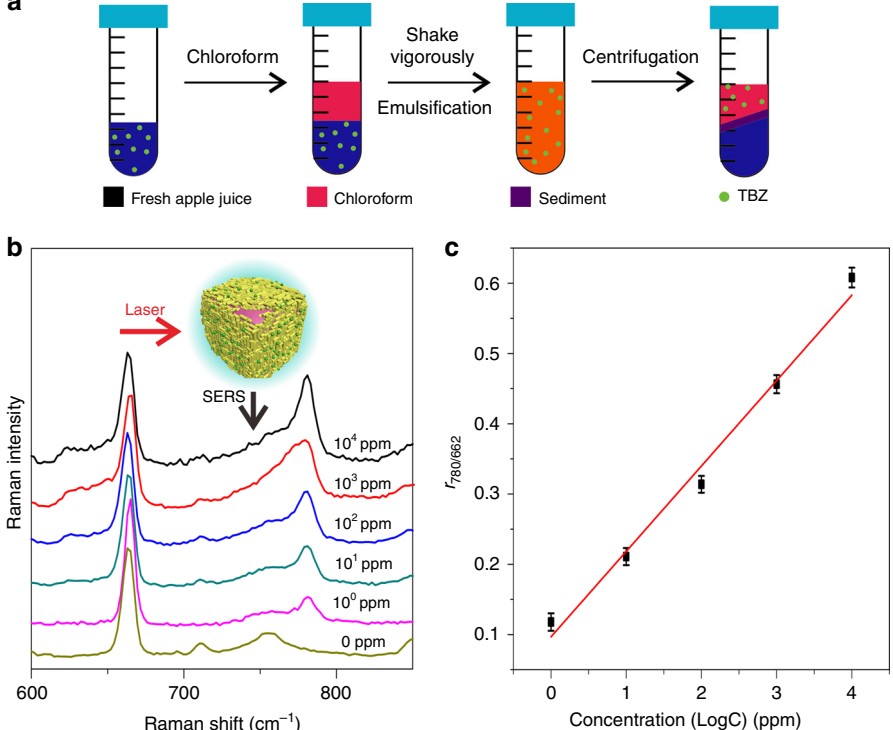

**Fig. 9** Quantitative SERS analysis of TBZ in fresh juice. **a** Schematic for separating and concentrating of TBZ from fresh apple juice. **b** SERS spectra of TBZ with concentrations of $10^0$, $10^1$, $10^2$, $10^3$, and $10^4$ ppm, respectively. **c** A linear plot of $r_{780/662}$ against logarithmic concentration of TBZ. The error bars represent the statistical RSD and were calculated from triplicate SERS experiments

might suffer from the interference of other impurities. Interestingly, chloroform could act as an efficient extraction agent to separate TBZ from the juice and enrich the TBZ molecules into the GNR nanogaps. After vortexing of Ci-GNR sols and the chloroform extract, the interfacial GNR arrays were directly used to generate corresponding SERS spectra. Clear fingerprint peaks of TBZ could be observed in the SERS spectra of each sample spiked with different TBZ concentrations. The band at 778 cm$^{-1}$ observed in pure TBZ solution shifted to 780 cm$^{-1}$ in real sample. This kind of variation could have originated from complex sample conditions. The $r_{780/662}$ value was plotted as a function of TBZ logarithmic concentration. A good linear relationship was achieved in the concentration range of $10^0$–$10^4$ ppm and reached LOD of 1 ppm for TBZ in juice, which meets the LOD requirement of 10 ppm in China's national food safety standard[45]. Both $O$-in-$W$ and the $W$-in-$O$ platforms exhibited excellent SERS sensitivity and reproducibility by the use of $O$ phase as an inherent IS tag (Table 1), and realized reliable SERS analysis of both Raman resonance dye molecules and non-resonance molecules that could be dissolved in either $O$ phase or $W$ phase at ultra-trace levels.

## Discussion
Typical solid substrates of GNRs were compared with our liquid interfacial GNR arrays (Supplementary Fig. 11). Our platform has much better sensitivity and reproducibility on detecting TBZ. It could be attributed to the poor solubility of TBZ in water and its weak affinity to fixed nanogaps of citrate-capped GNRs. In contrast, interfacial GNR arrays with dynamic nanogaps are flexible and self-healing, have tunable plasmonic hot spots, and allow free diffusion of surrounding molecules close to the nanogaps. Under $O/W$ biphasic conditions, analytes usually with dielectric behaviors can directly cooperate in the self-assembly, be efficiently localized by liquid interfacial tension, and realize self-

**Table 1 Performance of the quantitative SERS analyzer**

| Analytes | Dissolved phases | SERS platform | Quantitative indicators | LOD | $R^2$ |
|---|---|---|---|---|---|
| TBZ | $O$ | $O$-in-$W$ | $r_{780/662}$ | 0.1 ppm | 0.997 |
| TBZ | $O$ | $W$-in-$O$ | $r_{780/662}$ | 0.1 ppm | 0.978 |
| MG | $W$ | $O$-in-$W$ | $r_{1612/662}$ | 0.1 ppb | 0.960 |
| CV | $O$ | $O$-in-$W$ | $r_{1614/665}$ | 0.1 ppb | 0.968 |
| CV | $W$ | $O$-in-$W$ | $r_{1614/665}$ | 0.1 ppb | 0.975 |
| R6G | $O$ | $O$-in-$W$ | $r_{1504/665}$ | 0.1 ppb | 0.975 |
| R6G | $W$ | $O$-in-$W$ | $r_{1506/665}$ | 0.1 ppb | 0.989 |
| TBZ[a] | $O$ | $O$-in-$W$ | $r_{778/662}$ | 1.0 ppm | 0.983 |

[a]SERS analysis of the TBZ in fresh apple juice

enrichment in the nanogaps. More importantly, the tunable feature of liquid interfacial array enables a uniform distribution of analytes, thereby dramatically improving signal reproducibility.

Salt-induced aggregates of GNR sols were also compared with our liquid interfacial GNR arrays under similar conditions (Supplementary Fig. 12). Our platform has much higher SERS sensitivity, but seemingly worse reproducibility of SERS signals without IS calculation. The former has homogeneous dispersion in liquid, and once the laser focusing was fixed, the plasmonic hot spots in the excitation volume would have very small or even no fluctuations, leading to signal stability. This also explains the low sensitivity since this homogeneous system decreased the overall density of both particles and molecules in excitation volume. Nevertheless, the IS-calculated SERS intensity of our platform generated an RSD of 6.2% (Fig. 7d), i.e. interfacial GNR arrays could generate sensitive and quantitative SERS signal with IS calculation.

Considering multiple globules of metal liquid-like GNR arrays, rather than a single globule, in the assembled macroscopic droplet, the focal volume of the excitation laser would cover different microenvironments at different runs, which is bad for signal reproducibility. Moreover, the layer-by-layer assembly of both silver nanowires[46] and nanorods[47] evidenced the saturation of SERS signals beyond several layers of nanomaterials owing to limited skin depth and laser penetration issues. Analytes face a similar dilemma in that the much larger superficial area of multiple globules dilutes the density of the analytes. More seriously, multiple globules may need plenty of surface ligands to maintain their topological structure, but these ligands usually greatly hinder the entrance of targeted molecules into active sites, leading to lower SERS sensitivity. Hence, we conclude that a single globule of our interfacial GNR arrays provides a flat side surface for laser focusing, which should lead to better SERS enhancement and better repeatability than typical solid substrates and multiple globules.

In summary, we have developed metal liquid-like 3D plasmonic arrays on O/W interface, which can undergo reversible encasing as the wettability of the vessel surface changes. This platform has remarkable SERS capability to quantitatively detect multiple analytes in the multiphase system for both water-soluble and oil-soluble molecules. It evidences sub-ppb levels of sensitivity for Raman resonance molecules and sub-ppm levels for non-resonance Raman molecule TBZ, particularly in a fresh juice system. Compared to typical discrete IS tags fixed on the nanoparticle surface, this homogeneous O phase as IS tags provide stable and uniform signal feedback and avoid the problems of competing adsorption or dynamic exchange on a metal surface. Our study enables the SERS technique to be developed into a simple, practical, and quantitative liquid-state analyzer in the future. The protocol for fabricating this liquid-state plasmonic platform is generic and paves a way for the use of other nanomaterials to tailor plasmonic response at liquid–liquid interface, which holds potential for application in sensors, phase-boundary catalysis, interfacial events and etc.

## Methods

**Chemicals**. Sodium borohydride (NaBH$_4$) and hexadecyltrimethylammonium bromide (C$_{16}$H$_{33}$(CH$_3$)$_3$NBr, CTAB) were purchased from Aladdin. Sodium polystyrenesulfonate (Na-PSS, Mw = 70 kDa) was purchased from Sigma Aldrich. Chloroauric acid hydrate (HAuCl$_4$·H$_2$O) was supplied by Nanjing Chemical Reagent Co., Ltd. Silver nitrate (AgNO$_3$), ascorbic acid (C$_6$H$_8$O$_6$), sulfuric acid (H$_2$SO$_4$), hydrogen peroxide (H$_2$O$_2$), chloroform (CHCl$_3$), ethanol (C$_2$H$_6$O), and sodium citrate (C$_6$H$_5$Na$_3$O$_7$·2H$_2$O) were obtained from Sinopharm Chemical Reagent Co. Ltd. Thiabendazole (C$_{10}$H$_7$N$_3$S, TBZ, 98%) and triethoxy-1H,1H,2H,2H-tridecafluoro-n-octylsilane (C$_{14}$H$_{19}$F$_{13}$O$_3$Si, F13S) were purchased from Adamas Reagent Co., Ltd. Malachite green (C$_{23}$H$_{25}$ClN$_2$, MG), rhodamine 6G(C$_{28}$H$_{31}$N$_2$O$_3$Cl, R6G), and crystal violet (C$_{25}$H$_{30}$ClN$_3$, CV) were purchased from Shanghai Yuanye Biological Technology Co. Ltd. All solutions and dilutions were prepared using ultrapure water with a resistivity larger than 18.2 MΩ cm.

**Apparatus**. Characterizations were performed on a UV-2600 spectrometer, a Sirion 200 field-emission scanning electron microscope (FESEM), and a JEOL 2010 transmission electron microscope (TEM), respectively. DFM was carried out on an inverted microscope (IX71, Olympus) outfitted with a ×60 objective lens (NA = 0.7), a dark-field condenser (0.8 < NA < 0.92), a true-color digital camera (Olympus DP80, Japan), and a white light source (100 W halogen lamp). Synchrotron radiation small-angle X-ray scattering (SR-SAXS) was carried out on beamline BL19U2 at the Shanghai Synchrotron Radiation Facility (SSRF) with a beam spot of 200 μm × 400 μm and wavelength λ of 0.124 nm (10 keV). The distance between sample and detector was 2228.6 mm, and the scattering vector (q) range was 0.03–1.5 nm$^{-1}$. SERS experiments were conducted on a portable BWTek i-Raman® Plus (Model # BWS465-785S) equipped with a standard BCR100A accessory for liquid measuring in a common cuvette. The parameters included a laser line of 785 nm, a power of 30 mW, an exposure time of 8 s, one accumulation, and a laser spot of 100 μm in diameter. The laser excitation is located on to the side wall of the cuvette, which is in close proximity to the GNR arrays.

**Citrate-stabilized GNR sols**. Ct-GNR sols were synthetized by a classical seed-mediated method[31]. Then, the capping agent, CTAB, was converted into citrate through a poly(4-styrenesulfonic acid) (PSS)-mediated process (Fig. 2a)[30]. Briefly, Ct-GNR sols were subjected to three cycles of centrifugation and redispersion (C/R) in 0.15 wt% Na-PSS (Mw = 70 kDa) to deplete CTAB to trace levels. The Pss-GNRs were then subjected to two additional C/R cycles using 2 mM sodium citrate for exchange with PSS, yielding stable dispersion of Ci-GNR sols. The as-prepared Ci-GNR sols have an absorbance of 3.0 at approximately 700 nm, denoted as OD = 3.0. Different amounts of GNR sols used in the experiments were all concentrated to the OD values of 1 mL Ci-GNR sols through centrifugation ignoring the loss and nonlinear absorption.

**Metal liquid-like GNR arrays**. One milliliter of Ci-GNR sols and 1 mL of chloroform were added into a glass vessel with either hydroxylated or fluorosilylated surface treatment, and the mixture was vigorously shaken for 1 min. After allowing it to stand for 30 s, a metal liquid-like brilliant golden film was formed spontaneously on the O/W interface. When the vessel surface was subjected to hydroxylated treatment, the 3D GNR array formed a film encasing the bottom O phase, but formed a film encasing the upper W phase when the vessel surface was subjected to fluorosilylated treatment.

**Surface treatment of glass vessels**. Glass vessels, including quartz cuvettes and bottles, were immersed in a boiling piranha solution of 30% H$_2$O$_2$ and concentrated H$_2$SO$_4$ at a volume ratio of 3:7 for 1 h. The hydroxylated vessels were rinsed repeatedly with ultrapure water and then immersed in a 40 mM F13S solution in ethanol for 24 h, followed by rinsing with ethanol. The fluorosilylated vessels were finally dried in the dry box at 90 °C.

**Liquid-state SERS analysis of pesticide TBZ in fresh juice**. TBZ, an oil-soluble pesticide, in fresh juice could be rapidly separated and purified by a simple chloroform extraction. In brief, 1 mL of fresh juice and 10 μL TBZ with different concentrations were added into a 4 mL EP tube. Then, 1 mL of chloroform, as the extraction solvent, was added and mixed well by vigorous shaking. Finally, the mixture was centrifuged at 12,000 g for 5 min, and the whole process was completed in 10 min. The collected chloroform phase was directly added to an aliquot of Ci-GNR sols to prepare metal liquid-like GNR arrays in a cuvette. This cuvette was directly measured in a customized measuring accessory (Fig. 1) on a portable Raman device as simple as UV–Vis spectroscopic measurements.

## Data availability

All data supporting this study and its findings are available within the article and its Supplementary Information or from the corresponding author upon reasonable request.

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

## Acknowledgements

This study was financially supported by the NSFC grants (21874034 and U1632116), the Fundamental Research Funds for the Central Universities, Key Research and Development Project of Anhui Province (1704a07020067), Key Projects of Applied Basic Research of Hunan Province (2016JC2065), and the China Postdoctoral Science Foundation (2015M582322 and 2016T90748). Special thanks to the Open Project of State Key Laboratory of Chemo/Biosensing and Chemometrics at Hunan University. This work is also supported by NSFC grants (21521063), and by NIH GM R35 127130 and NSF 1645215.

## Author contributions

H.L. and W.T. conceived and designed the experiments. L.T., M.S., H.L., and W.T. wrote the manuscript. L.T. and M.S. conducted all the experiments. F.Y., Y.X., and L.L. assisted with GNR synthesis, SEM and DFM characterization. X.L. helps to do the SR-SAXS experiments and data analysis. All authors discussed the results and commented on the manuscript.

## Additional information

**Competing interests:** The authors declare no competing interests.

