## [Peer Review File · Nature Communications]

Reviewers' comments:

Reviewer #1 (Remarks to the Author):

This paper reports the formation of gold rod film at liquid interfaces (chloroform/water) for solid-substrate free, liquid-state SERS. The authors carefully present quantitative SERS measurements for TBZ as well as other dye molecules. I agree with the authors that liquid-state SERS platform in a common cuvette would be very useful in many practical sensing applications.

However, as the authors indicate in part in the manuscript, liquid-state SERS in a simple cuvette by taking advantage of the autonomous assembly and orientation of gold nanorod at oil/water interface (reference number 22) was already demonstrated in Nature Communications (Nature Communications, 4, 2182 (2013)) a few years ago. Interfacial SERS detection for multiphase trace molecule detection by utilizing the assembly of nanoparticles was also published in Nature Materials (Nature Materials, 12, 165 (2013)) (reference number 21).

On the basis of the above concerns, unfortunately, I do not think that the major claims of the paper are enough novel and interesting to a broad audience. Instead, I feel that this work become suitable for more specialized journals like Analytical Chemistry after addressing the following points.

1. The authors need to provide detailed characterization data on the gold nanorod film (e.g., the number of gold nanorod layer, orientation, relative distance):
2. SERS measurements of the aggregate of colloidal gold nanorods for the molecules the authors tested should be compared.
3. Why does CTAB limit the self-assembly of GNR arrays at liquid/liquid interface, and while citrate doesn't? Many previous studies have demonstrated the assembly of CTAB-capped GNRs at LLIs.
4. Why does the shape of the GNR become a dog bone shape in each step of replacement of capping agent?
5. The location at which Raman is measured should be specified.
6. The assemblies of nanoparticles at liquid/liquid interfaces are well known. What is the main advantage (particularly for SERS) of this assembly compared to other liquid/liquid interface assemblies?

Minor corrections;

1. In Figure 1A, citrate is written as "cirate".

2. The Raman spectra in Figure 7B seem to be labeled inversely.

Reviewer #2 (Remarks to the Author):

The authors describe a method of making 3D gold nanorods array on a chloroform / water interface that has attractive SERS capability. The system is able to undergo reverse encasing, depending on the wettability of the cuvette. The advantage of this system is that it is able to simultaneously detect analytes dissolved in both the O and W phases, the O phase also act as an extraction agent and inherent label. The work describe in this manuscript is quite different from the roughened metallic substrates and random metal nanoparticles aggregates in solution that are commonly reported in SERS publications. The authors also contrasted it with their previous work on 2D arrays on horizontal interfaces. In my opinion, this is an interesting piece of work that will be of interest to those who are developing SERS strategies.

The experimental approach is sound and the quality of the data and presentation is good. The detail of the experimental procedure provided is sufficient.

Here are some suggestions for the authors to strengthen their manuscript:

1. In this work, a single globule of either W or O is formed, with the GNR assembling at the interface. Will the system have better SERS sensitivity, if there are multiple globules instead since this will result in much larger surface array for assembly of the GNR?
2. I notice that the authors have a published paper on 3D SERS hotspots using assembled spherical colloidal superstructure (Analytical Chemistry, 2015, 87, 4821) which is not cited in the manuscript. The published work has some overlap with the current manuscript and needs to be discussed
3. The authors to comment if there is any strategy to further lower the standard deviation of the Raman signal in their platform. A standard deviation of less than 10% or even less than 5% will be necessary for practical quantitative analysis.
4. The authors should explain how is the nanogaps between the nanorods at the interphase controlled. Does the amount of nanorods added affect the resonance wavelength and SERS sensitivity? Is the nanogaps already optimized in this work, and if yes, how was it optimized? It is not clear if all the nanorods assembled at the interface or there's and excess that remains in the water phase.
5. The authors should do a fair discussion on how their system is advantageous to typical solid SERS substrates that is most commonly reported for SERS.
6. A video to show the forming of the O in W and transits to W and O system will be of interest

to readers.

Reviewer #3 (Remarks to the Author):

Liu present a very interesting paper on the detection of analytes using liquid liquid interface. Overall I believe there is sufficient novelty to merit publication; however, I have the following remarks:

1. The introduction needs to better emphasize the state of the art which includes putting the work in context of that already published by other groups (e.g. Girault, Kornyshev, Dryfe). Much of the most important work is not appropriately cited. Furthermore, a number of methods and procedures have been taken from the literature and appropriate citation would be relevant.
2. Fig 1. Hard to follow exactly how and where the particles are excited. Revising the diagram would be appropriate. It would also be beneficial if information such as laser wavelength could be incorporated.
3. metal liquid-like 3D GNR needs to be put in better context especially in relation to Girault's recent work.
4. There is limited discussion on surface functionality in terms of total charge groups, although this is hard to quantify it clearly plays a very important role in making of the films and would benefit from more in-depth discussion.
5. Figure 3d/4d/5d. the caption states that these are histograms; however, this does not appear to be the case and needs to be revised accordingly.
6. Purely a personal opinion; however, it is like much of figure 3-5 could be put in the SI with only the key outcomes being shown in the main text otherwise the discussion is a little repetitive.

Responses to the Editorial Office and Reviewers

Thanks to the Editorial Office and Reviewers for their recommendations and many thanks for their time and thoughtful comments and suggestions. It is very impressive that some similar questions were raised by the reviewers, which have been addressed together. The related materials have been incorporated into the revised manuscript and supplementary materials. The main revised parts of the manuscript are highlighted in RED.

Reviewer #1 (Remarks to the Author):

This paper reports the formation of gold rod film at liquid interfaces (chloroform/water) for solid-substrate free, liquid-state SERS. The authors carefully present quantitative SERS measurements for TBZ as well as other dye molecules. I agree with the authors that liquid-state SERS platform in a common cuvette would be very useful in many practical sensing applications.

However, as the authors indicate in part in the manuscript, liquid-state SERS in a simple cuvette by taking advantage of the autonomous assembly and orientation of gold nanorod at oil/water interface (reference number 22) was already demonstrated in Nature Communications () a few years ago. Interfacial SERS detection for multiphase trace molecule detection by utilizing the assembly of nanoparticles was also published in Nature Materials () (reference number 21).

On the basis of the above concerns, unfortunately, I do not think that the major claims of the paper are enough novel and interesting to a broad audience. Instead, I feel that this work become suitable for more specialized journals like Analytical Chemistry after addressing the following points.

Response: Thanks the reviewer for this insightful comment. It is certainly true that the innovations in our work must be compared with those contained in two seminal reports from Prof. Edel's group (*Nat. Mater.* 2013, 12, 165) and Prof. Kang's group (*Nat. Commun.* 2013, 4, 2182). The comparison was also the focus of our introduction in earlier manuscript although may not have been clear enough.

Very recently, Prof. Tian's group reviewed the bottlenecks and future directions of SERS. One direction involves fundamental research to extend the limits of SERS; in contrast, the other involves practical research to expand the range of applications with the aim of providing versatile analytical tools for surface, materials, life, environmental, forensic and food sciences, among others (Tian Z.Q.,* et al. *Chem. Commun.* 2018, 54, 10). The bottleneck of SERS impeding the development of a practical rapid analyzer is not about the Raman devices, but rather about the total solution from

SAMPLING to SPECTRUM. Both solid substrate-based and nanoparticle sol-based SERS sensors face the contradiction between sensitivity and reproducibility. Five liquid-state systems of Au sols may be used for SERS measurements (Scheme R1). The isolated particle sols or salt-induced aggregates have poor sensitivity as shown in Figure R5, and the concentrated sols have the problems of laser penetration length and reproducibility. In 2012, Girault and colleagues reported on the first experimental demonstration of mirror-like properties of 2D self-assembled monolayers of Au NPs at the liquid-liquid interface, implying an impressive capacity of this new kind of plasmonic platform for tunable optical devices, sensors, and catalysis.

In February of 2013, Prof. Edel’s group reported on the self-assembly of GNPs into close-packed arrays at liquid/liquid interfaces with 20 mM NaCl as promoter to increase the efficiency of assembly. Their SERS measurements were performed as follows: “...the sample was transferred onto a 130–160- μm -thick coverslip (Fig. 1a(v)), which resulted in the nanoparticles forming a thin film (in this case, the aqueous phase was below the nanoparticles and the organic phase above). The diameter of the O/W interface once placed on the coverslip was approximately 5 mm.”

Scheme R2. Adopted from Figure 1a-v in *Nat. Mater.* 2013, 12, 165 (Left) and Figure 1b in *Nat. Commun.* 2013, 4, 2182 (Right).

In June of 2013, Prof. Kang’s group reported on the self-orientation of CTAB-capped GNRs at oleic acid–water interface to create a substrate-free interfacial liquid-state SERS. Clearly, these two

reports performed SERS on a 2D horizontal planar array of gold nanoparticles. This involves a very fundamental tradeoff between precise laser focusing and light-path arrangement. Both of the above platforms face fluctuations in liquid levels, especially for applications in portable Raman devices. Furthermore, the later GNRs platform adopts a transmission-type signal collection of SERS. As a result, the optical path-length might be a limiting factor in the performance of this platform based on the relatively weak signals of R6G in Figure 4h and high concentrations of targets used in Figure 5. In addition, CTAB, as a surface-capping agent, may also hinder targets close to the metal surface. Nevertheless, these two excellent works demonstrate a versatile interfacial liquid-state SERS technique and significantly expands the flexibility of SERS.

Scheme R3. Our reversible O/W encasing platform for SERS analysis.

Developing a miniaturized, easy-to-prepare 3D volumetric liquid-state interfacial platform promises a higher capability that would allow plasmonic arrays to be excited more efficiently and more easily. In addition, liquid-state interfacial SERS promises a practical, substrate-free, and rapid analysis, but still faces a great challenge in developing a batch and uniform fabrication strategy with stable internal standards (IS) owing to the difficulties in precisely locating both the IS tags and analytes in the same local structure under the harsh conditions of biphasic liquid interface. The enhancement effect always fluctuates because of the sampling and measuring environment, making quantitative analysis challenging. Hence, developing a liquid-state and quantitative SERS platform in a common cuvette is very attractive. Our study is a solid step in this direction.

More specifically, our study demonstrates a 3D liquid interfacial self-assembly of GNR arrays with features of self-healing, robustness, reproducibility, and fabrication without any inducers or modifiers. The rapid self-assembly of GNRs at the O/W interface produced a metal liquid-like

brilliant golden droplet. This 3D volumetric interfacial array is very stable and free from fluctuation of liquid levels, which is more convenient for measurements made with a portable device. More interestingly, a reversible chloroform/water (O/W) encasing in a cuvette was realized, just by the surface wettability of the container, and showed attractive quantitative capability of SERS signals. This is a significant contribution to the SERS detection of various targets with different natures. Furthermore, the O phase, chloroform itself, generated a stable SERS fingerprint peak and was used as an inherent IS for quantitative SERS analysis. Both O-in-W and W-in-O platforms exhibited excellent SERS sensitivity and reproducibility on a portable Raman device with a laser line of 785 nm. Reliable SERS quantitation was realized for various Raman resonance and non-resonance molecules, and targeted analytes dissolved in both O and W phases could also be distinguished at sub-ppb levels. Hence, our study paves the way toward practical and quantitative liquid-state SERS analysis in a common cuvette as simple as a UV-Vis spectrometer.

We think our study is not only a simple SERS analytical strategy, but also a solid step toward practicality in implementing the SERS technique, when compared to previous frameworks. This practical and quantitative tool promises wide applications in sensors, phase-boundary catalysis, liquid interfacial molecular events, and even interfacial reaction mechanisms. We therefore believe that our contribution could be of interest to the broad readership of *Nat. Commun.* Thanks again to the reviewer for this thoughtful review.

Q1. The authors need to provide detailed characterization data on the gold nanorod film (e.g., the number of gold nanorod layer, orientation, relative distance)

Response: The exact structure of layers of nanoparticles at the liquid-liquid interface (O/W interface) is extremely difficult to assess (*Nat. Mater.* 2013, 12, 165). Here, we first employed dark-field microscopy to observe the assembly morphology on O/W interface and scanning electron microscopy was used to evidence the nature of the dried assembly array on silicon wafer.

Figure R1. (A) Schematic diagram of dark-field microscopy (DFM). (B) *In situ* DFM observation of metal liquid-like O-in-W GNR arrays in PDMS cavity with 1 mL of 0.6 OD GNR sols, and the scale bar is 4 μm . (C) SEM observations of close-packed GNR film dried on silicon wafer fabricated from 1 mL of 7.0 OD GNR sols, and the scale bar is 1 μm .

GNR orientation was first investigated in Prof. Kang's report which used the linear polarization of incident light in DFM to demonstrate vertical orientation of CTAB-capped GNRs at the oleic acid–water interface. We did not have the experimental conditions for this experiment. Nevertheless, we collected conventional DFM image of GNRs on the O/W interface (Figure R1A). The image displays discrete particles with various colors, but red is dominant (Figure R1B). The results indicate a predominant longitudinal surface plasmon resonance band of GNR on the O/W interface, whereas most GNRs have horizontal orientation on the O/W interface, although this does not exclude the influence of nanorod aggregates. This inference is consistent with SEM observations of close-packed GNR film dried on a silicon wafer (Figure R1C). Assuming GNR arrays had vertical orientation on O/W interface under close-packed conditions, the dried array should have vertical GNRs or at least fall down in preferential direction. Of course, SEM images did not produce data that would lead to the conclusion that GNR arrays had vertical orientation. Hence, we consider GNR on O/W interface to have a horizontal orientation, which is also the lowest energy state of GNRs on O/W interface.

The number of GNR layers on O/W interface was further observed. At low concentration of GNRs, the array arrangement is single layer and has big gaps (red arrowheads in Figure S3). As the number of GNRs increases, the interparticle spaces gradually decrease. At higher concentration, the

GNR arrangement appears multilayered, likely a result of solvent evaporation, and the interparticle spaces could not be further decreased because of the energy barriers at O/W interface where excess GNPs tend to be arranged in multilayers. This inference is consistent with the optical observations that the single-layer or subsingle-layer assembly of GNPs on O/W interface produces a metal liquid-like brilliant golden droplet, but that much higher concentrations cause surface wrinkling and loss of metallic luster.

Figure R2. (A-H) SEM images of metal liquid-like arrays fabricated by 1 mL GNR sols with different OD values: 1.0, 2.2, 3.4, 4.6, 5.8, 7.0, 8.2, 9.4, and the scale bar is 1 μm . (I-L) DFM images of diluted interfacial arrays fabricated by 1 mL GNR sols with different OD values: 0.3, 0.6, 0.9, 1.2, and the scale bar is 20 μm . (M-P) Enlarged details of the corresponding DFM images in I-L, and the scale bar is 5 μm .

Figure R3. (A) A typical two-dimensional SR-SAXS pattern. (B) One-dimensional SR-SAXS scattering data and fitting lines of GNR sols and interfacial GNR arrays, respectively, plotted as SR-SAXS intensity (I) vs. scattering vector modulus (q). (C) Transformed SR-SAXS curves in B, plotted as $q^4 \cdot I$ vs. q . (D) Calculated average particle size (*meansize*) and center-to-center distance (*ETA*) from different samples: 1, as-synthesized GNR sols of 3 OD; 2-4, interfacial arrays fabricated by GNR sols with variable OD values: 3.0, 6.0, 9.0, respectively.

Figure R4. Calculated particle size distribution of four different samples in Figure R3D.

More importantly, *in situ* synchrotron-radiation small-angle X-ray scattering (SR-SAXS) measurements were performed to get information on the orientation and relative distance of GNRs array at the O/W interface. A typical 2D SR-SAXS pattern (Figure R3A) was integrated with 1D

SR-SAXS scattering curves (Figure R3B), and the scattering intensity I of the SAXS curves was transformed into $q^4 \cdot I$ plotted as a function of q to highlight the scattering peaks of GNRs (Figure R3C). According to the scattering spectrum, we can get the GNRs arrangement at the O/W interface.

Assuming a cylindrical model of GNR with a length-to-diameter ratio of 3.0 based on TEM images, the formula $F(q)$ is the form factor of cylindrical particles as

$$F(q, \beta, r) = 2V\rho_0 \frac{\sin(qH\cos\beta)}{qH\cos\beta} \times \frac{J_1(qR\sin\beta)}{qR\sin\beta}, \quad (3)$$

where H is the height of the cylinder, R is the radius, β is the orientation angle, and our sample is random where q is the scattering wave vector, and $J_1(x)$ is a first order Bessel function. ETA is the center-to-center distance between particles, and $meansize$ is the average scattering size of particles. The calculated ETA and $meansize$ (particle size) values are plotted as functions of volume of GNRs by the use of Irena software (Figure 4D). The results show that GNRs at O/W interface have a $meansize$ of 47.8 nm, which is consistent with TEM observations, and that the distribution of particle sizes (Figure S4) implies the discrete states of GNRs rather than the aggregates after self-assembly on O/W interface. In addition, the ETA values gradually decreased from 85.4 nm to 26.1 nm, i.e., the interparticle space decreased down to negative values at larger GNR densities.

Under limited conditions, it should be noted that ETA values are a fitting parameter for which the tendency toward relative changes is more important than numerical magnitudes. As mentioned above, high concentrations of GNRs at O/W interface did not induce interparticle aggregation; hence, the decreased ETA values at high GNR concentrations could be attributed to the multilayer arrangements of GNRs at O/W interface. The negative values of interparticle space can be explained by the effect of GNRs in multilayer arrangements which would have no interparticle space for X-ray penetration in physical. The results, when combined with SEM and DFM observations, indicated that close-packed arrays of GNR multilayers were formed under high concentrations of GNRs.

Q2. SERS measurements of the aggregate of colloidal gold nanorods for the molecules the authors tested should be compared.

Response: Thanks for the suggestions. Here, the typical platform of salt-induced aggregates of GNR sols was compared with our metal liquid-like interfacial GNR arrays. Moreover, in response to Q5 of Reviewer 2, we have also performed a comparison between our system and typical solid substrates most commonly reported for SERS. One mL of 6.0 OD GNR sols was used for both platforms. The

O-in-W GNR array was prepared as described in the manuscript. The aggregate of GNR sols was produced by adding 0.5 M of sodium chloride and resulted in a homogeneous as-prepared aggregating system, showing a deep-blue color, that could stabilize within a half hour.

The analyte TBZ had a final concentration of 10 ppm in O-in-W interfacial GNR arrays, but 100 ppm in salt-induced aggregates of GNR sols. Moreover, the excitation laser was 60 mW for interfacial GNR arrays but 90 mW for aggregated GNR sols. To explain the difference, the latter platform could not generate SERS signals with strong enough intensities at lower concentrations of TBZ. In other words, the aggregated GNR sols system used 10 times higher TBZ concentration and 1.5 times higher laser power, but only generated rather weak SERS signals approximately 10% of that obtained from GNR arrays at O/W interface, indicating that interfacial GNR arrays have much higher SERS sensitivity.

Figure R5. Comparison between (A&C) metal liquid-like GNR arrays (O-in-W) and (B&D) salt-induced aggregates of GNR sols with respect to UV-Vis absorbance, optical topography, and SERS intensity histograms at 780 cm^{-1} (I_{780}) without IS calculations. Two platforms used the same concentrations of the GNR sols, but the former had the analyte TBZ of 100 ppm dissolved in ethanol.

Interestingly, as shown in Figure R5C and D, no SERS intensities at 780 cm^{-1} were calculated with IS tags, and while the former platform generated an RSD of 12.8% in SERS intensity, the latter platform only generated 8.3%. Seemingly, the salt-induced aggregates of GNR sols have better reproducibility of SERS signals, most likely because the salt-induced aggregates of GNR sols have homogeneous dispersion in liquid phase. Once having fixed the focus plane and excitation volume,

the plasmonic hotspots in the excitation volume would have very small, or even no, fluctuations, leading to signal stability. This also explains the low sensitivity since this homogeneous system decreased the number of hotspots and molecules in excitation volume. In contrast, the GNRs at O/W interface form dense arrays which results in the extreme concentration of SERS hot spots. Targeted molecules are also enriched in the interface, generating much higher SERS sensitivity. Moreover, the O phase, as internal standard, could well calibrate the SERS signals, and the IS-calculated SERS intensity generated an RSD of 6.2% (Figure 6D). Hence, our platform could generate sensitive and quantitative SERS signal with the 3D interfacial self-assembly and IS calculation.

Q3. Why does CTAB limit the self-assembly of GNR arrays at liquid/liquid interface, and while citrate doesn't? Many previous studies have demonstrated the assembly of CTAB-capped GNRs at O/W interfaces. (**Q4 of Reviewer 3.** There is limited discussion on surface functionality in terms of total charge groups, although this is hard to quantify it clearly plays a very important role in making of the films and would benefit from more in-depth discussion.)

Response: Thanks to the reviewers for these two similar suggestions. We do not deny the capability of CTAB for self-assembly of GNRs at liquid/liquid interfaces. In fact, some studies in the literature have reported on the aforementioned self-assembly of GNRs (*Nat. Commun.* 2013, 4, 2182; *JPN. J. APPL. PHYS.* 2004, 43, 554-556). However, to the best of our knowledge, the self-assembly of CTAB-capped GNRs only involves 2D horizontal arrays on planar liquid/liquid interface, and CTAB-assisted assembly of GNRs were mostly transferred through LB membrane technology to solid surface for subsequent SERS detection.

In the context of surface chemistry, CTAB is a cationic surfactant, and CTAB-capped GNRs were positively charged, thus promoting the self-assembly of nanoparticles. Nonetheless, this behavior appears to be concentration-dependent, making the assembly morphology and interparticle gap hard to control as a consequence of the uncertain layer number of CTAB micelle (*Langmuir*. 2005, 21, 2923-2929; *Nano Lett.* 2011, 11, 5013-5019). High concentrations of CTAB would contaminate ligand-modified GNRs and interfere with Raman enhancement.

We also tried to produce metal liquid-like interfacial arrays by using CTAB-capped GNRs. Emulsification occurred after CTAB-capped GNR sol was oscillated by vortex for a period of time. However, the emulsions could not fuse into a single globule, possibly because the bilayer micelle of CTAB molecules increased steric hindrance, thereby impeding GNR accumulation onto the O/W

interface.. In response to Q1 of Reviewer 2, we found that Ag nanowires together with gold nanospheres could also form a similar single globule with the capping agent of citrate, but excess ligand PVP used to synthesize Ag nanowire solution would impede this assembly into a single globule. Hence, the surfactants that could form bilayer micelles must be removed, or reduced, in the assembly of single globule.

Hence, we speculate that the assembly of the nanoparticles might rely on the spontaneous diffusion-limited nanoparticle localization to the O/W interface, as well as an increased efficiency of assembly with increasing ionic strength of the aqueous phase. The emulsification process has a key role in reducing the average distance between the nanoparticles and the O/W interface, thereby speeding up diffusion-limited localization to the interface (*Nat. Mater.* 2013, 12, 165-171). Additionally, as mentioned by Girault and the colleagues, vigorous shaking, or, alternatively, ultrasonication, effectively facilitates the spontaneous adsorption of individual nanoparticles at the interface (*ACS Nano* 2014, 8, 9471-9481). The driving force behind this spontaneous interfacial adsorption is the diminution of excess surface energy at an early stage of the citrate-capped GNRs. Citrate-capped GNR sols provide a “cleaner” plasmonic surface, and vortex mixing creates shear forces that can be particularly strong at W/O interfaces, facilitating GNRs to reach the liquid boundary and localize onto the O/W interface because of O/W interfacial tension. Moreover, interparticle van der Waals forces likely contribute significantly to the formation of GNR film on the O/W interface.

Q4. Why does the shape of the GNR become a dog bone shape in each step of replacement of capping agent?

Response: Conventional synthesis of GNR sols could not repeatedly produce 100% regular GNR every time. Instead, a significant fraction of thermodynamically favorable irregular structures, such as dog bone-like or dumbbell-shaped structures, are frequently formed (*RSC. Adv.*, 2016, 6, 30028-30036). A common reason for this failure is the slight change in the surrounding temperature or duration of growth, which is almost inevitable. The main reason for forming dog bone-like GNRs is the preferential binding of CTAB molecules to the middle of the nanorods, resulting in the deposition of more Au atoms at the ends (*Chem. Soc. Rev.* 2013, 42, 2679-2724). Ligand exchange is important for forming an ordered structure, but this is usually problematic for CTAB-capped GNRs. CTAB, a surfactant used in the synthesis of GNRs, can be removed by multiple washes and

exchanges with chemisorptive surfactants, phospholipids, or other surface-active agents. However, ligand exchange has addressed these difficulties by providing procedures for high-yield ligand exchange products that effectively remove CTAB. GNRs often aggregate irreversibly during ligand exchange from excessive destabilization of the CTAB bilayer before the GNR surface is effectively passivated with the incoming ligand (*Adv. Mater.* 2012, 24, 4811-4841). On the other hand, multi-step encapsulation with polymers increases the thickness that precludes a small-molecule analyte from accessing electric field enhancement. Ligand exchange is predicated on employing a molecule able to displace CTAB from the GNR surface to allow favorable binding to the GNR surface and ensure its tight binding to the Au surface under different solution conditions (*Nano Lett.* 2011, 11, 5013–5019). During the process of displacing CTAB by other ligands in our experiments, it should be noted that prolonged centrifugation and redispersion might alter the surface structure of GNRs to some extent. Also, the uneven capping of CTAB and subsequent polymer or citrate could induce atomic rearrangements of GNRs and produce irregular structures, such as the aforementioned dogbone-like shape.

As shown in SEM of Figure R1C above, *Ci*-GNR with fairly good uniformity can be successfully fabricated by carefully and systematically controlling room temperature and pH during the seed-mediated growth approach. More important is whether regularly cylindrical or dog bone-like GNRs can easily realize the self-assembly of metal liquid-like GNR array. In addition, different sizes of citrate-capped Ag NPs, Ag nanowires and Au NPs have also been used for successful interfacial self-assembly of metal liquid-like array (data not shown), and this work is ongoing in our group.

Q5. The location at which Raman is measured should be specified. (Q2 of Reviewer 3. Fig 1. Hard to follow exactly how and where the particles are excited. Revising the diagram would be appropriate. It would also be beneficial if information such as laser wavelength could be incorporated.)

Response: Figure 1 has been revised according to your suggestions. As shown in Scheme 1 and Figure R6, the as-prepared GNR array was a metal liquid-like macroscopic droplet, and this platform was a self-healing, robust, reproducible, and reversible encasing determined just by the surface wettability of the container. Moreover, the volume of both O and W phases could be reduced or removed. This step could increase particle density at the O/W interface and could also adjust the

position of this droplet in the container. Laser excitation is located on the side wall of the cuvette, which is in close proximity to the GNR arrays. The detailed illustrations and experimental setups were as follows:

Figure R6. Illustrations on reversible O/W encasing for self-assembling metal liquid-like GNR array as a multiphase liquid-state SERS analyzer (Left) and the experimental setups of GNR array excited by fiber-optics probe on hand-held Raman device (Right).

Q6. The assemblies of nanoparticles at liquid/liquid interface are well known. What is the main advantage (particularly for SERS) of this assembly compared to other liquid/liquid interface assemblies?

Response: In response, we have discussed above our innovations compared with previous reports. Here, we briefly discuss the main advantages of our platform for SERS technique.

We have developed a reversible chloroform/water (O/W) encasing strategy to self-assemble metal liquid-like 3D GNR arrays, just by controlling the surface wettability of the containers. (1) Citrate-capped GNR (*Ci*-GNR) can quickly self-assemble at the O/W interface into large-scale metal liquid-like 3D droplets. (2) This process is triggered by the addition of chloroform into the GNR sols without any inducers or modifiers. *Ci*-GNRs ensure a “cleaner” plasmonic surface for easier closing of targets to metal surface than other surfactants. (3) The O phase itself generates a stable SERS fingerprint peak that can be used as a good IS tag for quantitative SERS analysis. By using a portable Raman device, the ratiometric SERS intensity (r) of target molecules was generated in a stable, continuous, and predictable manner with the signals of O phase as the normalization standard. (4) Chloroform could act as an extraction solvent, possessing higher extraction rates of many oil-soluble molecules with less interference from impurities. Targeted analytes from both organic and aqueous

phases could be well distinguished.

In brief, our platform is self-healing, easy to operate, and it needs no engineering. Its good uniformity promises excellent SERS performance, indicating high potential for the rapid detection of analytes dispersed in multiplex and multiphase systems. We think our study is not only a simple SERS analytical strategy, but also a solid step toward implementing a practical SERS technique on the basis of previous frameworks. Our study paves the way toward practical and quantitative liquid-state SERS analysis in a common cuvette as simple as a UV-Vis spectrometer. This practical and quantitative tool promises wide applications in sensors, phase-boundary catalysis, liquid interfacial molecular events, and even interfacial reaction mechanisms.

Minor corrections:

1. In Figure 1A, citrate is written as “cirate”.

Response: Done.

2. The Raman spectra in Figure 7B seem to be labeled inversely.

Response: Figure 7B was corrected.

Reviewer #2

The authors describe a method of making 3D gold nanorods array on a chloroform / water interface that has attractive SERS capability. The system is able to undergo reverse encasing, depending on the wettability of the cuvette. The advantage of this system is that it is able to simultaneously detect analytes dissolved in both the O and W phases, the O phase also act as an extraction agent and inherent label. The work describe in this manuscript is quite different from the roughened metallic substrates and random metal nanoparticles aggregates in solution that are commonly reported in SERS publications. The authors also contrasted it with their previous work on 2D arrays on horizontal interfaces. In my opinion, this is an interesting piece of work that will be of interest to those who are developing SERS strategies.

The experimental approach is sound and the quality of the data and presentation is good. The detail of the experimental procedure provided is sufficient.

Here are some suggestions for the authors to strengthen their manuscript:

Q1. In this work, a single globule of either W or O is formed, with the GNR assembling at the interface. Will the system have better SERS sensitivity, if there are multiple globules instead since this will result in much larger surface array for assembly of the GNR?

Response: Thanks to the reviewer for these thoughtful comments. Actually, it is a great challenge for us in a short time to construct such a system of multiple globules under conditions similar to the single globule. For example, in response to Q3 of Reviewer 1, emulsification occurred after CTAB-capped GNR sol was oscillated by vortex for a period of time; however, the emulsions could not fuse into a single globule, possibly caused by the bilayer micelle of CTAB molecules increasing steric hindrance and hindering GNR accumulation onto the O/W interface. Furthermore, we have tried many surfactants and nanoparticles, which could either easily form a single globule or form many tiny emulsions that would not contribute to good SERS performance.

Accidentally, we found that PVP-capped Ag nanowires together with citrate-capped gold nanospheres could form such stable system of multiple globules (Figure R7). When PVP was changed to citrate through a multiple centrifugation and redispersion process, citrate-capped Ag NWs together with Au NPs could easily form a single globule. The determinant factor is just the amount of residual PVP molecules in the final system. In detail, the newly PVP-synthesized Ag NWs and citrate-capped Au NPs are mixed in equal volumes and mixed with equal aliquot of chloroform containing 100 ppm TBZ to form multiple globules by shaking. As a control, the newly PVP-synthesized Ag NWs were washed three times by centrifugation to remove excess PVP, and similar procedures resulted in a single globule. We can conclude from this experiment that excess ligand PVP used to synthesize the Ag nanowire solution will form bilayer micelles and hinder self-assembly into a single globule. These two hybrid assemblies of Ag NWs and Au NPs were used as model systems to compare SERS capability between a single large globule and multiple small globules.

Figure R7C and D represent 2D SERS spectral mappings of different runs on a single globule and multiple globules, respectively. Two main peaks could be observed. One at 662 cm^{-1} can be attributed to chloroform molecules, and the other at 780 cm^{-1} can be attributed to TBZ molecules. It could be easily found that the single globule generated much higher relative intensity of 780 cm^{-1} compared to 662 cm^{-1} . Moreover, the single globule exhibits much better repeatability of SERS intensity at 780 cm^{-1} compared to multiple globules, as shown in Figure R7E and F.

When multiple globules of different sizes exist in a cuvette, the focal volume of the excitation laser will cover different microenvironments at different runs. For example, the laser may focus on the gap between two globules or the side of single globule. This is bad for signal repeatability. It should be noted that the Raman enhancement of multilayer films does not always increase along with the number of layers. For instance, a layer-by-layer assembly of an Ag nanowire woodpile structure

for 3D SERS application evidenced the saturation of SERS beyond three layers of Ag nanowires owing to limited skin depth and laser penetration issues (Langmuir 2013, 29, 7061-7069). Very recently, similar results were observed on layer-by-layer assembly of Ag nanorod substrates (Anal. Chem. 90, 12, 7275-7282). This occurred because some hot spots in 3D architectures are embedded by other surface NPs and thus do not contribute to the final SERS enhancement. Analytes face a similar dilemma in that the much larger superficial area of multiple globules dilutes the density of the analytes.

More seriously, multiple globules need plenty of surface ligands to maintain their topological structure. These ligands would greatly hinder the entrance of targeted molecules into active sites, leading to lower SERS intensity. In contrast, when a single globule is formed, the laser focuses on a flat surface. The measured signal shows good reproducibility at different runs because of the good self-healing properties. Here, although we use Ag NWs and Au NPs to construct the model system, we believe that the conclusion is general. A single globule will produce better SERS enhancement and better repeatability than multiple globules.

The discussion on single and multiple globules has been incorporated into the revised manuscript, and the related data is ongoing work in our group. Therefore, Figure R7 was not put into the Supporting Information and is herewith submitted for your kind review only.

Q2. I notice that the authors have a published paper on 3D SERS hotspots using assembled spherical colloidal superstructure (Analytical Chemistry, 2015, 87, 4821) which is not cited in the manuscript. The published work has some overlap with the current manuscript and needs to be discussed.

Response: This work has been cited in the revised manuscript. In our previous work, an emulsion template method was used to assemble the oleylamine-capped Ag NPs into 3D colloidal superstructures in a controlled way via CYH/water biphasic mixing. However, it should be noted that the 3D colloidal superstructures formed only after the complete evaporation of the CYH phase, and “finally, the products, 3D colloidal superstructures, were dispersed in 2 mL of ultrapure water.” SERS measurement in this previous report was as claimed: “1 μ L of the above products (namely Ag colloidal superstructure) was dropped on a silicon slide and dried in air at room temperature. 1 μ L of analyte solution was dropped on the dried film of the above products and then used for SERS analysis.” i.e., dried film of 3D Ag superstructures acts as SERS substrates, but not liquid-state SERS measurements. We speculate that liquid-state SERS measurements on this system would face

difficulties similar to those listed in Q1 of Reviewer 1, i.e., many tiny emulsions would not contribute to good SERS performance.

The current work focused on self-assembly of GNRs at the chloroform/water interface into large-scale metal liquid-like 3D droplets without any inducers or modifiers and realized the arbitrary transition of oil-in-water or water-in-oil state. Subsequently, the liquid interface array contained analyte already localized to hotspots during the GNRs self-assembly process and was directly detected and analyzed. The superiority of the liquid interface platform lies in its self-healing, mechanical flexibility and defect-free pristine nature (*ACS Nano*. 2013, 7, 9526-9532). Also, it can reduce sample damage owing to laser heating effects. Furthermore, chloroform itself generated a stable SERS fingerprint peak that could be used as a useful internal standard for quantitative SERS analysis.

Q3. The authors to comment if there is any strategy to further lower the standard deviation of the Raman signal in their platform. A standard deviation of less than 10% or even less than 5% will be necessary for practical quantitative analysis. **Q4.** The authors should explain how is the nanogaps between the nanorods at the interphase controlled. Dose the amount of nanoroad added affect the resonance wavelength and SERS sensitivity? Is the nanogaps already optimized in this work, and if yes, how was it optimized? it is not clear if all the nanorods assembled at the interface or there's and excess that remains in the water phase.

Response: We answer the two questions together. The nanogaps between GNRs have been optimized by controlling the amount of GNR sols added in the system. The comprehensive experimental data of SEM, DFM and SR-SAXS reveal the controllability of nanogaps between GNRs (Figure R1-4).

Figure R8. (A) Optical images of as-synthesized GNR sols and metal liquid-like GNR arrays on O-in-W interface fabricated by 1 mL of GNR sols with varied OD values: 3.0, 4.5, 6.0, 7.5, 9.0, respectively. (B) The corresponding UV-Vis absorbance spectra, (C) Relative SERS strength, $r_{780/662}$, collected on interfacial GNR arrays, and (D) statistical histograms of $r_{780/662}$ and I_{780} , respectively.

Figure R9. (A) UV-Vis absorbance spectra of the aqueous phase before and after interfacial assembly with 1 mL of 7.5 OD GNR sols. (B) DFM observations on the edge of metal liquid-like droplet fabricated with 0.6 OD GNRs, and the scale bar is 20 μm.

To optimize the SERS sensing performance, a series GNR sol concentrations was used to produce the self-assembled metal liquid-like GNR arrays (Figure 6A). We found that UV absorption strength of metal liquid-like GNR arrays increased with the increase of GNR concentrations and that the absorbance maximum of interfacial GNR arrays gradually redshifted from 680 nm to 800 nm (Figure 6B) when the OD values of 1 mL GNR sols increased from 3.0 to 9.0, implying a gradually

enhanced interparticle SPR coupling. When the concentration of 1 mL GNR sols was 7.5 OD, almost no GNRs were left in the aqueous solution after interfacial self-assembly (Figure S10A). A DFM image was obtained at diluted GNR sols, but it still clearly shows that nearly all GNRs are concentrated on the O/W interface (Figure S10B). Based on SEM images and SR-SAXS data, it was concluded that a closely packed monolayer of GNR array will form at the GNR concentration of approximately 7.0 OD.

TBZ of 10 ppm was then used to examine SERS performance. Figure 6C showed that SERS signals first increased quickly with increasing GNR concentrations and then reached a plateau with GNR sols of 7.5 OD, even slightly decreasing at higher GNR concentrations. However, when the concentration of GNP sols was approximately 7.0 OD, the enhancement effect of SERS was the best. Fifty runs in triplicate SERS experiments generated an RSD of $r_{780/662}$ and I_{780} with values of 6.2% and 12.8%, respectively (Figure 6D). These results demonstrate that IS tags could greatly improve signal reproducibility. The $r_{780/662}$ at 7.5 OD GNR sols was 1.63, which is 4.3 times greater than that at 3.0 OD GNR sols. Thus, by optimizing the amount of GNR added, SERS sensitivity was greatly improved. Nevertheless, with any further increase in GNR concentration, the residual GNR in W phase might adsorb part of the analytes and induce a decrease in SERS intensity.

Q5. The authors should do a fair discussion on how their system is advantageous to typical solid SERS substrates that is most commonly reported for SERS.

Response: Thanks for the suggestions. Here, the O/W interface platform was compared with fixed GNR arrays dried on Si wafer. This question was also raised by Reviewer 1 (Q2), and we responded with a summary of our comparison between the typical platform of salt-induced aggregates of GNR sols and our metal liquid-like interfacial GNR arrays.

Figure R10. (A) The SERS spectrum of TBZ with different concentration on the Si wafer solid platform, (B) Forty random runs in triplicate SERS experiments generating a 2D spectral mapping of 100 ppm TBZ. (C) Statistical histograms of I_{780} collected on Si wafer solid platform.

As shown in Figure R10, both O/W interfacial arrays and solid film produced a similar detection limit of TBZ, i.e. 0.05 ppm. With the same laser excitation, SERS intensity of 10 ppm TBZ in O/W interfacial GNR arrays exceeded 8000 counts, but 100 ppm of TBZ dropped on the solid substrate only generated SERS intensity of approximately 1500 counts. Meanwhile, the signal-to-noise ratio of SERS spectra on the solid substrate was much worse than that of the liquid interface. Figure S12B shows 2D spectral mapping; only one dominant peak at 780 cm^{-1} was observed. The RSD of O/W interfacial GNR arrays was greatly improved, as shown in Figure S12C, compared to that of solid substrate. Usually, fixed solid nanostructures of noble metals on silicon wafers will produce high sensitivity, even if poor reproducibility was often reported. Therefore, we asked why solid substrate, herein described, would result in such poor sensitivity.

To address this question, we considered two factors that can affect SERS detection of TBZ on solid substrate. One involves the molecular affinity of TBZ on a metal surface. Solid substrates here are randomly aggregated GNRs, and the interparticle nanogap is fixed. TBZ is an oil-soluble molecule without resonance Raman effects; as such, it has poor affinity to the citrate-capped GNR surface. Another factor is the solubility of TBZ molecules, which are only soluble in organic solvents, such as chloroform and ethanol. Such organic solvents would quickly spread on the solid surface once dropped on to solid films of GNRs, decreasing the molecular density. But O/W interfacial GNR

arrays are flexible, self-healing and have tunable plasmonic hotspots. TBZ molecules can easily enter the hotspot region. Interfacial GNR arrays with dynamic nanogaps allow free diffusion of surrounding molecules close to the nanogaps. Under O/W biphasic conditions, analytes might directly take part in the self-assembly process. They can be efficiently adsorbed into the nanogaps of GNR arrays by interfacial tension and achieve self-enrichment. More importantly, the feature of interfacial array enables even adsorption of molecules in arrays, resulting in a more uniform distribution of molecules at the liquid interface, thereby achieving a dramatic improvement of signal reproducibility.

Q6. A video to show the forming of the O in W and transits to W and O system will be of interest to readers.

Response: It is a good suggestion. A video on three cycles of reversible O/W transition of metal liquid-like GNR arrays has been supplied as supporting material.

Reviewer #3

Liu present a very interesting paper on the detection of analytes using liquid liquid interface. Overall I believe there is sufficient novelty to merit publication; however, I have the following remarks:

Q1. The introduction needs to better emphasize the state of the art which includes putting the work in context of that already published by other groups (e.g. Girault, Kornyshev, Dryfe). Much of the most important work is not appropriately cited. Furthermore, a number of methods and procedures have been taken from the literature and appropriate citation would be relevant.

Response: This is a very meaningful suggestion. In recent years, many techniques and methods have been proposed for the liquid-liquid interface assembly of noble metal nanoparticles. Self-assembled nanoparticle (NP) arrays at liquid interfaces provide a unique optical response which has opened the door to new tunable metamaterials for sensing and optical applications. The related progress has been incorporated into the introduction of revised manuscript. The related context is as follows:

“Girault et al. reported the first experimental demonstration of mirror-like properties of two-dimensional (2D) horizontal arrays of gold nanoparticles at the liquid-liquid interface,¹⁹ implying an impressive capacity of this new kind of plasmonic platform for tunable optical devices,²⁰ sensors,²¹ and catalysis.^{22,23} These platforms have also been used to achieve SERS detection of multiple analytes from the aqueous, organic, or air phases with high sensitivity.²⁴⁻²⁶ Compared to the “fixed” solid array and the random aggregates in sols, 2D interfacial liquid-state array provides a

molecularly sharp and defect-free focal plane.²⁷ Specifically, it does not need engineering, can self-assemble if the proper conditions are provided, and such arrays are variable, versatile and self-healing, virtually guaranteeing the feasibility of localizing analytes to nanogaps and the stable controlling of gap size.^{25,28} However, SERS measurements on a 2D horizontal planar array involves a very fundamental tradeoff between precise laser focusing and light-path arrangement,^{26,29} which encounters fluctuations in liquid levels, especially for portable Raman devices, and transmission-type signal collection, which limits SERS performance. Recently, Girault et al. introduced the assembly of reflective liquid-like gold that encases macroscopic droplets, denoted as gold metal liquid-like droplets,³⁰ with the assistance of the lipophilic electron donor tetrathiafulvalene, an organosulfur compound, under emulsified conditions.

Usually, the self-assembly of nanoparticles at the liquid-liquid interface requires the assistance of salts, inducers, promoters, or accelerators for structural aggregating,^{31,32} or altering the surface charges of nanoparticles,^{33,34} and reducing the Coulomb repulsion between the nanoparticles.³⁵ Our previous work³⁶ developed an emulsion template method to assemble oleylamine-capped Ag NPs into 3D colloidal superstructures in a controlled way via CYH/water biphasic mixing. Acting as SERS substrates, dried film of 3D Ag superstructures would face similar difficulties. However, liquid-state tiny emulsions of this type would not result in good SERS performance. Complex inducers or surfactants might also hinder targets close to the metal surface. Therefore, developing a miniaturized, readily prepared 3D liquid-state plasmonic platform for quantitative SERS analysis in a common cuvette would promise a higher capability than 2D horizontal planar array and dried film and allow more efficient and easier excitation.³⁷ Such device would pave the way toward practical and quantitative SERS measurement in a common cuvette, which could be likened to a simple UV-Vis spectrometer, and it would be far superior to typical solid substrate-based or nanoparticle sol-based analysis.”

Q2. Fig 1. Hard to follow exactly how and where the particles are excited. Revising the diagram would be appropriate. It would also be beneficial if information such as laser wavelength could be incorporated.

Response: Done! The diagram has been revised, and the information of laser wavelength has been incorporated as shown in the response to Q5 of Reviewer 1.

Q3. Metal liquid-like 3D GNR needs to be put in better context especially in relation to Girault’s

recent work.

Response: Done! As mentioned in the response to Q1 of Reviewer 3, the concept and development of metal liquid-like plasmonic platform has been incorporated into the context of the revised introduction.

Q4. There is limited discussion on surface functionality in terms of total charge groups, although this is hard to quantify it clearly plays a very important role in making of the films and would benefit from more in-depth discussion.

Response: Done. Please review the response to Q3 Reviewer 1.

Q5. Figure 3d/4d/5d. the caption states that these are histograms; however, this does not appear to be the case and needs to be revised accordingly.

Response: The label and caption of Figure 3d/4d/5d have been revised.

Q6. Purely a personal opinion; however, it is like much of figure3-5 could be put in the SI with only the key outcomes being shown in the main text otherwise the discussion is a little repetitive.

Response: Done. Figure 4 and 5 have been relocated to SI as Figure S7 and S8, and new data on SR-SAXS, dark-field microscopy, and SEM have been added as new figures in the revised manuscript.

Reviewers' Comments:

Reviewer #1 (Remarks to the Author):

During the revision, the authors addressed in full all technical comments I raised. I would recommend the publication of this work in Nature Communications.

Reviewer #2 (Remarks to the Author):

The authors have put in commendable effort in their reply to the questions raised by the reviewers. They have performed additional experiments to support their response. I am satisfied with their point to point response and will recommend acceptance of the manuscript.

Reviewer #3 (Remarks to the Author):

To my opinion the authors responded to comments of all referees as much as we could at the current state of experiments. I have no further objections against publication of this paper.